# The Lie of the Average: How Class Incremental Learning Evaluation Deceives You?

**Guannan Lai**[1,2]    **Da-Wei Zhou**[1,2]    **Xin Yang**[3]    **Han-Jia Ye**[1,2]*

[1] School of Artificial Intelligence, Nanjing University
[2] National Key Laboratory for Novel Software Technology, Nanjing University
[3] School of Computing and Artificial Intelligence, Southwestern University of Finance and Economics
*Email: {laign, zhoudw, yehj}@lamda.nju.edu.cn, yangxin@swufe.edu.cn*

## Abstract

Class Incremental Learning (CIL) requires models to continuously learn new classes without forgetting previously learned ones, while maintaining stable performance across all possible class sequences. In real-world settings, the order in which classes arrive is diverse and unpredictable, and model performance can vary substantially across different sequences. Yet mainstream evaluation protocols calculate mean and variance from only a small set of randomly sampled sequences. Our theoretical analysis and empirical results demonstrate that this sampling strategy fails to capture the full performance range, resulting in biased mean estimates and a severe underestimation of the true variance in the performance distribution. We therefore contend that a robust CIL evaluation protocol should accurately characterize and estimate the entire performance distribution. To this end, we introduce the concept of extreme sequences and provide theoretical justification for their crucial role in the reliable evaluation of CIL. Moreover, we observe a consistent positive correlation between inter-task similarity and model performance, a relation that can be leveraged to guide the search for extreme sequences. Building on these insights, we propose **EDGE** (Extreme case–based Distribution & Generalization Evaluation), an evaluation protocol that adaptively identifies and samples extreme class sequences using inter-task similarity, offering a closer approximation of the ground-truth performance distribution. Extensive experiments demonstrate that EDGE effectively captures performance extremes and yields more accurate estimates of distributional boundaries, providing actionable insights for model selection and robustness checking. Our code is available at https://github.com/AIGNLAI/EDGE.

## 1 Introduction

Class Incremental Learning (CIL) seeks to equip a model with the ability to incorporate new class knowledge over time while preserving accurate recall of previously learned classes (Zhou et al., 2024a,b; Li et al., 2025). While much of the literature has centered on advancing architectures and algorithms, the equally crucial question of **how we evaluate CIL** has received far less attention. Recent studies reveal that final performance in CIL is highly sensitive to the sequence in which new classes arrive (Bell & Lawrence, 2022; Lin et al., 2023; Shan et al., 2024; Wu et al., 2021). Such sensitivity to class order is particularly problematic in realistic settings (e.g., autonomous driving), where the order of class emergence is inherently uncontrollable. Compounding this challenge, the number of possible sequences grows factorially with the number of classes ($O(N!)$), rendering exhaustive evaluation impractical. Consequently, CIL evaluation must rely on sampling only a subset of class sequences to assess and compare model performance.

Existing CIL evaluation protocols (Wang et al., 2024b; Zhou et al., 2024b) typically compute model capability by sampling only 3-5 random class sequences and reporting the sample mean and standard deviation — an approach we call the **Random Sampling (RS) protocol**. Because RS relies on only a handful of sequences, it yields only point estimates and provides no characterization of the full

---
*Corresponding author.

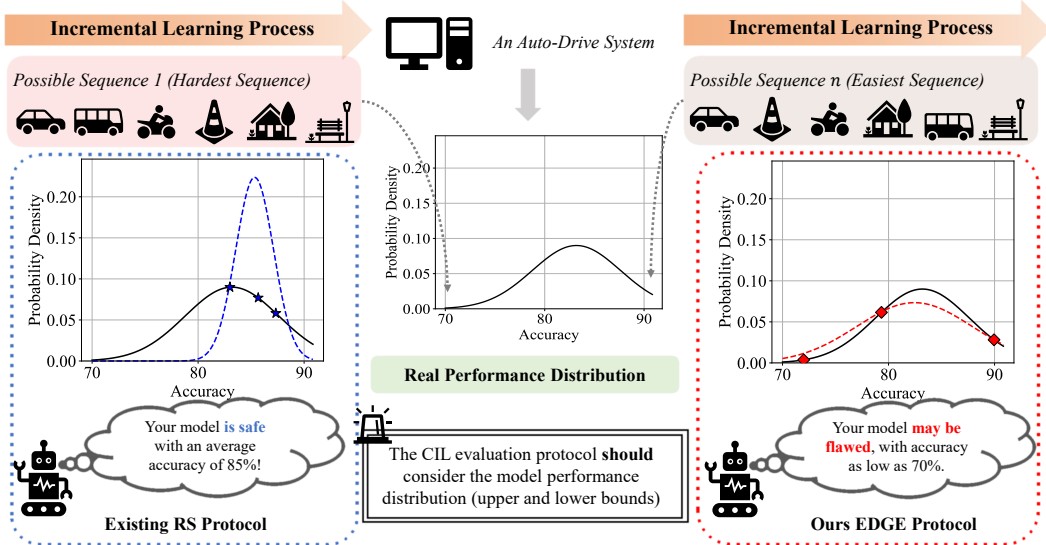

Figure 1: Existing CIL evaluations may be misleading! They merely compute the average accuracy without perceiving the performance distribution, failing to anticipate the impact of potential extreme sequences on the model.

performance distribution. To examine whether RS can reliably approximate the true performance distribution, we conduct a controlled study with **6** classes organized into **3** sequential tasks, resulting in **90** possible class-arrival orders. We exhaustively evaluate each sequence to obtain the ground-truth distribution of test accuracies. Figure 1 illustrates this experimental setting. Consider a realistic incremental-training scenario, such as an autonomous driving system, where numerous intrinsically different class-arrival sequences may occur in practice. Evaluating all sequences in our controlled study produces the ground-truth performance distribution. Analysis of this distribution reveals two key observations: first, the model performance approximately follows a Gaussian shape; second, there is substantial variation in extreme cases, with the gap between the easiest and hardest sequences reaching up to **20%** in our example (Hide-Prompt (Wang et al., 2023b) on CIFAR-100 (Krizhevsky, 2009)).

Following the RS protocol, we emulate typical practice by randomly sampling three sequences and fitting a Gaussian $\mathcal{N}(\mu, \sigma^2)$ using their sample mean and variance. As highlighted in the blue box of Figure 1, comparing this RS-estimated Gaussian against the ground-truth distribution reveals systematic bias: RS tends to *overestimate the mean*, *dramatically underestimate the variance*, and fails to capture the true upper and lower performance bounds. Consequently, selecting models based solely on the reported average is risky — a model with an inflated mean but a poor lower bound may cause severe failures in real-world deployment. These observations demonstrate that RS is inadequate for faithfully capturing CIL performance; a reliable evaluation protocol must either characterize distributional extremes or otherwise provide a substantially better approximation of the full performance distribution.

Motivated by the RS protocol's neglect of extreme sequences and supported by our theoretical analysis of these cases, we adopt extreme case sampling to more comprehensively characterize model capability and achieve more accurate performance estimates. Through both theoretical and empirical analysis of inter-task similarity and model performance, we identify inter-task similarity as a key factor influencing CIL performance. Building on this insight, we propose **EDGE** (*Extreme case-based Distribution & Generalization Evaluation*), a novel evaluation framework for CIL. EDGE encodes class-level textual descriptions using a pre-trained CLIP model to construct a class similarity matrix. It then generates three representative class sequences: one that maximizes inter-task similarity to simulate an *easy* scenario, one that minimizes it to represent a *difficult* scenario, and one randomly sampled to serve as a *medium* case. Model performance is evaluated on these three sequences, and their results are aggregated by computing the mean and standard deviation, providing a more comprehensive approximation of the model's performance distribution. As highlighted in the red box of Figure 1, EDGE produces a substantially closer approximation to the ground-truth distribution than the RS protocol, capturing both the central tendency and the distributional extremes.

The main contributions of this work are summarized as follows:

- We conduct a systematic study of evaluation protocols in CIL, emphasizing that evaluation should aim to capture the full performance distribution of a model. Through both theoretical analysis and empirical investigation, we show that the widely adopted RS protocol produces biased estimates and fails to reflect the realistic behavior of CIL models.
- We propose **EDGE** (*Extreme case-based Distribution & Generalization Evaluation*), a novel evaluation framework that adaptively identifies and samples both easy and challenging class sequences based on inter-task similarity, thereby providing a more faithful approximation of the ground-truth performance distribution.
- Extensive experiments validate the effectiveness of EDGE in sampling extreme sequences and estimating model performance accurately. Our analysis also uncovers notable phenomena, such as different methods exhibiting comparable lower-bound performance in specific scenarios, offering critical insights for the design of future CIL models.

## 2 RELATED WORK

**Class Incremental Learning (CIL):** Existing CIL approaches can be broadly categorized into non-pre-trained and pre-trained based methods (Cao et al., 2023; Dohare et al., 2024). Non-pre-trained methods typically fall into three categories: (1) Regularization-based methods, which introduce explicit regularization terms into the loss function to balance the learning dynamics between old and new tasks (Aljundi et al., 2018; Kirkpatrick et al., 2017; Li & Hoiem, 2017; Wang et al., 2022c); (2) Replay-based methods, which alleviate catastrophic forgetting by replaying data from past tasks, either through stored exemplars (Cha et al., 2021; Lopez-Paz & Ranzato, 2017; Riemer et al., 2018; Wang et al., 2022a) or via generative samples synthesized by GANs (Cong et al., 2020; Liu et al., 2020; Shin et al., 2017; Zhu et al., 2022); and (3) Dynamic network methods, which modify the network architecture—such as by expanding layers or neurons—to accommodate new knowledge while preserving prior information (Aljundi et al., 2017; Cao et al., 2025; Ostapenko et al., 2021; Wang et al., 2023c, 2022b). In contrast, PTM-based methods leverage the representational power of pre-trained backbones and mitigate forgetting through three main strategies: (1) Prompt-based methods, which apply lightweight updates via prompt tuning while freezing the backbone to maintain generalization (Jia et al., 2022; Li et al., 2024; Smith et al., 2023; Wang et al., 2023a, 2022d,e); (2) Model mixture–based methods, which store intermediate checkpoints and integrate them using ensemble or model-merging techniques (Gao et al., 2023; Wang et al., 2024a, 2023d; Zheng et al., 2023; Zhou et al., 2024c, 2025); and (3) Prototype-based methods, which classify examples using nearest-class-mean strategies grounded in PTM-derived embeddings (Lai et al., 2025; McDonnell et al., 2024; Panos et al., 2023; Zhou et al., 2024a).

**Evaluation Protocols of CIL:** Evaluation protocols in CIL have received comparatively limited attention. Prior studies such as Farquhar & Gal (2018); Hsu et al. (2018); Mundt et al. (2021) propose multi-dimensional assessment criteria and benchmarks, while Chen et al. (2025) investigates dynamic task allocation to probe lower-bound performance. In contrast, our work adopts a distribution-oriented perspective: rather than relying on a few random trials, we aim to estimate the underlying performance distribution via a small set of informative, extreme-aware samples. This approach enables more reliable assessment under atypical or adversarial class sequences and delivers more actionable guidance for model selection and design.

## 3 PRELIMINARIES

### 3.1 PROBLEM DEFINITION

**Class Incremental Learning (CIL).** Given an ordered sequence of tasks $\{1, \ldots, t, \ldots\}$, each task $i$ is associated with a training set $\mathcal{D}^i = \{X^i, Y^i\}$, where $X^i$ denotes the input samples and $Y^i$ the corresponding labels. Let $CLS^i$ denote the class set for task $i$, with cardinality $|CLS^i|$. A crucial constraint enforces strict separation between tasks: $\forall i \neq j \in \{1, \ldots, n\}$, $CLS^i \cap CLS^j = \emptyset$, and no inter-task data accessibility is allowed during training. The goal of CIL is to learn a unified embedding function $\Psi : \mathcal{D}^i \to \mathbb{R}^d$ that maps inputs to a shared embedding space, along with a classifier $f(\cdot)$ capable of maintaining discriminative performance across all encountered tasks.

In the CIL scenario, given a sequence of learning classes $\mathcal{O}$ comprising $T$ tasks, we define $A_{t,t'}$ as the classification accuracy of the model on the test set of the $t'$-th task after training on the first $t$ tasks. Based on this, the overall evaluation metric for sequence $\mathcal{O}$ can be formally defined as:

$$\mathcal{A}(\mathcal{O}) = \frac{1}{T} \sum_{t=1}^{T} A_{T,t}. \tag{1}$$

**Objective of CIL Evaluation Protocol Design.** Let $\Omega$ denote the space of all possible class sequences under a given CIL setting. By sampling $L$ sequences $\{\mathcal{O}_1, \ldots, \mathcal{O}_L\} \subset \Omega$ and computing their final accuracies $\mathcal{A}(\mathcal{O}_l)$ using Equation (1), we construct an empirical performance distribution $\mathcal{P}_{\text{emp}}$ with mean $\mu_{\mathcal{A}}$ and standard deviation $\sigma_{\mathcal{A}}$.

As shown in Figure 1 and the *appendix*, the realistic distribution $\mathcal{P}_{\text{true}}$ is approximately Gaussian. We therefore use $\mathcal{N}(\mu_{\mathcal{A}}, \sigma_{\mathcal{A}}^2)$ to approximate it, and define the goal of protocol design as minimizing the distributional distance between $\mathcal{P}_{\text{true}}$ and its estimate, measured by metrics such as Jensen-Shannon divergence (Lamberti et al., 2007) or Wasserstein distance (Villani, 2009).

**Random Sampling (RS) Evaluation Protocol.** For a given CIL model $\mathcal{M}$, the conventional evaluation protocol uses three fixed random seeds (Lai et al., 2025; Li & Zhou, 2025; McDonnell et al., 2024; Wang et al., 2022e) to generate class sequences $\{RS_l\}_{l=1}^3$. The performance of the model is then estimated by computing the mean and standard deviation of final accuracies: $\mu_{\mathcal{A}} = \frac{1}{3} \sum_{l=1}^{3} \mathcal{A}(RS_l)$, $\sigma_{\mathcal{A}}^2 = \frac{1}{3} \sum_{l=1}^{3} (\mathcal{A}(RS_l) - \mu_{\mathcal{A}})^2$. However, prior work only uses these statistics to summarize performance, without evaluating how well the estimated distribution matches the true one. This leads to overconfidence in the evaluation results and may result in misleading conclusions.

## 3.2 Limitations of RS Evaluation Protocol

Despite the substantial advances in CIL, to our knowledge, no prior work has critically examined the validity of prevailing evaluation protocols. In this section, we undertake theoretical investigations to address this oversight. *All theoretical results and proofs in this section are provided in the Appendix.*

First, we demonstrate through Lemma 1 that CIL evaluation cannot be reliably accomplished using existing RS protocols, due to the combinatorial explosion in the number of possible class sequences.

**Lemma 1.** *Let $N$ be the total number of classes, partitioned into $K$ tasks of equal size $M = N/K$. Then the number of distinct class sequences is $|\Omega| = \frac{N!}{(M!)^K}$. Moreover, under linear scaling $K = \Theta(N)$, the quantity $|\Omega|$ grows factorially, satisfying $|\Omega| = \Omega\big((N/e)^N\big)$, which asymptotically dwarfs any polynomial-scale sampling capacity as $N \to \infty$.*

For $N = 100$ classes divided into $K = 10$ tasks, the number of possible sequences is approximately $10^{92}$, vastly exceeding practical enumeration. The RS protocols typically sample only 3 class sequences, covering less than $10^{-90}\%$ of the space and thus suffering from severe under-sampling bias. Building on Lemma 1, we now ask: ***How Many*** random sequence samples are required to approximate the true accuracy distribution over the full sequence space within a given tolerance?

**Theorem 1.** *Let $\Omega$ be the set of all possible class sequences with $|\Omega|$ elements, and fix tolerance $\varepsilon > 0$ and failure probability $\delta \in (0, 1)$. Suppose we draw $L$ sequences $\{RS_l\}_{l=1}^L$ without replacement uniformly from $\Omega$, and let $\widehat{\mathcal{A}}_L = \frac{1}{L} \sum_{l=1}^{L} \mathcal{A}(RS_l)$ be the empirical mean, respectively. Then for any $\varepsilon > 0$, if*

$$L \frac{|\Omega| - L}{|\Omega| - 1} \geq \frac{\ln\big(2\,|\Omega|/\delta\big)}{2\,\varepsilon^2}, \tag{2}$$

*then with probability at least $1 - \delta$, $\big|\widehat{\mathcal{A}}_L - \mathbb{E}_\omega[\mathcal{A}(\omega)]\big| \leq \varepsilon$.*

**Remark 1.** *Substituting $|\Omega| = N!/(M!)^K \approx (N/e)^N$ from Lemma 1, the condition equation 2 becomes*

$$L \frac{\frac{N!}{(M!)^K} - L}{\frac{N!}{(M!)^K} - 1} \geq \frac{\ln\big(2/\delta\big) + \ln(N!/(M!)^K)}{2\,\varepsilon^2} \approx \frac{1}{2\varepsilon^2}\Big[N \ln(N/e) + \ln\frac{2}{\delta}\Big]. \tag{3}$$

*For large $|\Omega|$ and $L \ll |\Omega|$, the finite-population correction $\frac{|\Omega| - L}{|\Omega| - 1} \approx 1$, so one recovers the same sample complexity scale $\Omega\big(\frac{N \ln N}{\varepsilon^2}\big)$ as in the with-replacement case. Even for moderate $N$ (e.g.*

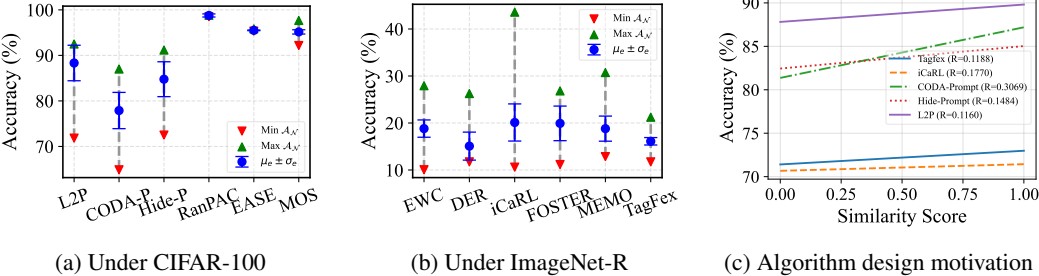

(a) Under CIFAR-100      (b) Under ImageNet-R      (c) Algorithm design motivation

Figure 2: Figures 2a and 2b show model performance under fully enumerable scenarios (green: maximum, red: minimum), along with estimates from the random sampling (RS) protocol (blue error bars). Figure 2c illustrates the correlation between inter-task similarity scores and model performance, where $R$ denotes the Pearson correlation coefficient.

$N = 100$) and a coarse $\varepsilon = 0.1$, achieving high confidence (say $\delta = 0.05$) still requires on the order of $L \gtrsim 2 \times 10^4$ samples, so purely random sampling remains fundamentally impractical.

Noting that in Equation (3) we have $L \ll |\Omega|$, let $E_t = \{ \omega \in \Omega : |\mathcal{A}(\omega) - \mathbb{E}_\omega[\mathcal{A}(\omega)]| > t \sqrt{\mathrm{Var}_\omega[\mathcal{A}(\omega)]} \}$. The probability that none of the $L$ sampled sequences falls into $E_t$ is approximately $\exp\big(-(|E_t|/|\Omega|)\,L\big)$. Hence, uniform random sampling almost surely fails to capture model performance in the most extreme cases. This observation motivates the idea of deliberately constructing such **extreme class sequences** to directly evaluate easy and hard case performance; Theorem 2 provides an initial theoretical analysis of this approach:

**Theorem 2.** *Let $\Omega$ be the set of all class sequences, and define $\mu = \mathbb{E}_{\omega \sim \Omega}[\mathcal{A}(\omega)]$, $\sigma = \sqrt{\mathrm{Var}_{\omega \sim \Omega}[\mathcal{A}(\omega)]}$ as the realistic mean and standard deviation of the accuracy function $\mathcal{A}$. Suppose we know two extreme sequences $\omega_+$, $\omega_-$ satisfying $\mathcal{A}(\omega_+) - \mu \geq \sigma$ and $\mu - \mathcal{A}(\omega_-) \geq \sigma$. Draw $L$ sequences $\{RS_l\}_{l=1}^L$ without replacement uniformly from $\Omega \setminus \{\omega_+, \omega_-\}$, and define $\widetilde{\mathcal{A}}_{L+2} = \frac{1}{L+2}\Big[\mathcal{A}(\omega_-) + \mathcal{A}(\omega_+) + \sum_{l=1}^L \mathcal{A}(RS_l)\Big]$. Then for any $\varepsilon > 0$ and $\delta \in (0,1)$, if*

$$L\,\frac{|\Omega| - 2 - L}{|\Omega| - 3} \geq \frac{\ln\big(2\,(|\Omega| - 2)/\delta\big)\,\big(R^{(\sigma)}\big)^2}{2\,\varepsilon^2}, \tag{4}$$

*where $R^{(\sigma)} = \mathcal{A}(\omega_+) - \mathcal{A}(\omega_-)$, then with probability at least $1 - \delta$, $\big|\widetilde{\mathcal{A}}_{L+2} - \mathbb{E}_{\omega \sim \Omega}[\mathcal{A}(\omega)]\big| \leq \varepsilon$.*

**Remark 2.** *Theorem 2 demonstrates that, under the conditions outlined in Remark 1, incorporating extreme class sequences reduces the required sample size to a value proportional to $\big(R^{(\sigma)}\big)^2$. For instance, when $R^{(\sigma)} \approx 0.1$ (which is common in practical scenarios), the lower bound on the sample size $L$ drops to around **50**. This represents a significant reduction compared to uniform random sampling, underscoring the practical benefit of extreme-sequence-assisted evaluation in CIL.*

## 4 EDGE: EXTREME CASE-BASED DISTRIBUTION & GENERALIZATION EVALUATION

### 4.1 MOTIVATION

Building on the theoretical analyses in Section 3.2, we conduct an exhaustive evaluation under a 6-class, 3-task setting. As illustrated in Figures 2a and 2b, the RS protocol often fails to accurately estimate the true performance distribution, frequently leading to either underestimation or overestimation of certain models, which compromises fairness in comparison. Meanwhile, the findings from Theorem 2, together with the **near-Gaussian** nature of the true distribution, highlight the importance of incorporating extreme class sequences to improve evaluation quality. Nevertheless, a key challenge remains in how to effectively leverage dataset-specific structures and characteristics to generate extreme sequences that are both robust and generalizable, thereby enabling more reliable and informative evaluation protocols.

In the CIL setting, it is intuitively understood that when adjacent tasks exhibit low similarity, model parameters undergo significant changes during task transitions, which increases the risk of forgetting.

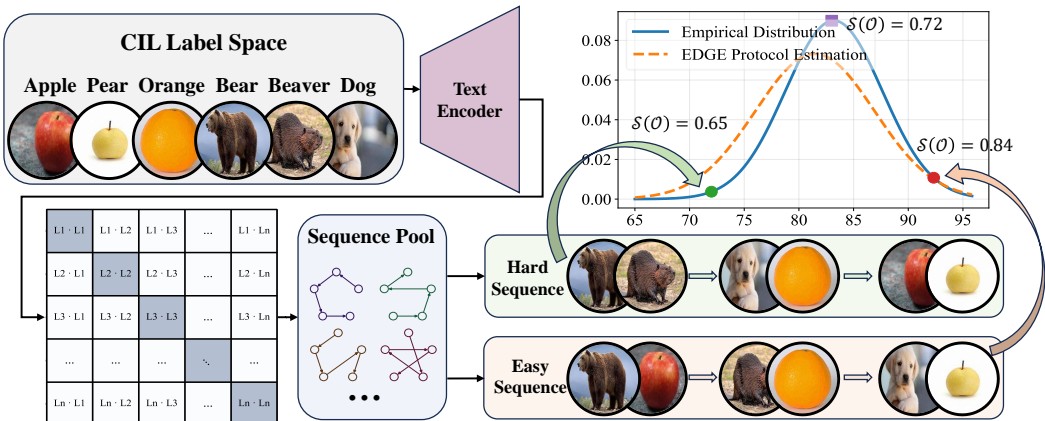

Figure 3: Illustration of the EDGE evaluation protocol. The sequence with a green background represents a hard case, where similar classes (e.g., apples and pears) appear within the same task, resulting in low inter-task similarity. The sequence with an orange background represents an easy case, where similar classes are distributed across different tasks, leading to high inter-task similarity.

To investigate this phenomenon further, we examine the relationship between inter-task similarity and model generalization error.

**Theorem 3.** *Consider a CIL system consisting of $K$ tasks, where each task $T_k$ is associated with a data distribution $\mathcal{D}_k$ and a class set $\mathcal{C}_k$. The generalization error is defined as $\epsilon_g = \frac{1}{K}\sum_{k=1}^{K} \mathbb{E}_{(x,y)\sim\mathcal{D}_k}[L(h(x),y)]$, where $L(h(x),y)$ denotes the loss between the model prediction $h(x)$ and the true label $y$. Given a task order $\mathcal{O} = \{T_1, T_2, \ldots, T_K\}$, the similarity score $\mathcal{S}(\mathcal{O})$ is defined as:*

$$\mathcal{S}(\mathcal{O}) = \frac{K}{(K-1)N} \sum_{1\leq i \leq K-1} \sum_{c\in\mathcal{C}_i} \sum_{c'\in\mathcal{C}_{i+1}} Sim(c,c'), \tag{5}$$

*where $Sim(c,c')$ denotes the semantic similarity in the representation space between classes $c$ and $c'$, belonging to tasks $T_i$ and $T_j$, respectively. Let $\mathcal{O}_h$ and $\mathcal{O}_e$ denote the sequences with the minimum and maximum similarity scores $\mathcal{S}(\mathcal{O})$, respectively, and let $\mathcal{O}_r$ represent a randomly generated sequence. Then, the following conditions hold:*

- *The similarity score satisfies $\mathcal{S}(\mathcal{O}_h) \leq \mathcal{S}(\mathcal{O}_r) \leq \mathcal{S}(\mathcal{O}_e)$,*

- *The generalization error satisfy $\epsilon_g(\mathcal{O}_h) \geq \epsilon_g(\mathcal{O}_r) \geq \epsilon_g(\mathcal{O}_e)$.*

Theorem 3 theoretically demonstrates that as task similarity decreases, the upper bound of the generalization error increases significantly. Figure 2c illustrates the trend between inter-task similarity scores and corresponding model performance for all possible class sequences. The majority of methods show a positive correlation, empirically supporting this result by showing a consistent decline in model accuracy as task similarity decreases. Motivated by these observations, we take advantage of inter-task similarity to construct extreme class sequences, which facilitates a more thorough and representative evaluation of CIL.

## 4.2 EXTREME SEQUENCE GENERATION ALGORITHM AND PROPOSED PROTOCOL

Figure 3 illustrates the proposed EDGE evaluation protocol. Given a dataset, since direct access to image instances is unavailable, we leverage the text encoder from a pre-trained CLIP model to embed class labels. Specifically, each class label is encoded into a $d$-dimensional semantic feature vector via the CLIP text encoder $\Phi$, forming a label feature set $\mathcal{L} = \{\mathbf{L}_1, \ldots, \mathbf{L}_N\}$, where $\mathbf{L}_i = \Phi(y_i) \in \mathbb{R}^d$. By computing cosine similarities between these label features, we construct a symmetric similarity matrix $\mathbf{D} \in \mathbb{R}^{N\times N}$, where each entry $d_{ij} = \frac{\mathbf{L}_i\cdot\mathbf{L}_j}{\|\mathbf{L}_i\|\cdot\|\mathbf{L}_j\|}$ quantifies the semantic similarity between classes $i$ and $j$. Based on the similarity matrix $\mathbf{D}$, we generate candidate class sequences by hierarchically clustering (Nielsen, 2016) semantically similar classes and strategically selecting subsequent tasks to minimize or maximize inter-task similarity. Two optimal permutations are identified using Equation (5): $\mathcal{O}_h = \arg\min_{o\in\Omega} \mathcal{S}(o)$ for the hardest sequence and $\mathcal{O}_e = \arg\max_{o\in\Omega} \mathcal{S}(o)$ for the

Table 1: Experimental results of pre-trained-based methods on two datasets. The gray region indicates the ground-truth values, and the best results are highlighted in bold black.

| Metric | CIFAR-100 | | | | | | ImageNet-R | | | | | |
|---|---|---|---|---|---|---|---|---|---|---|---|---|
| | L2P | CODA-Prompt | Hide-Prompt | EASE | MOS | RanPAC | L2P | CODA-Prompt | Hide-Prompt | EASE | MOS | RanPAC |
| $\min_{\mathcal{A}_\mathcal{N}}$ | 71.83 | 64.67 | 72.50 | 95.33 | 92.17 | 98.50 | 57.75 | 18.72 | 65.90 | 88.24 | 85.56 | 90.91 |
| RS | 83.83 | 76.83 | 79.67 | 95.50 | 95.33 | **98.67** | 68.98 | 39.04 | 78.34 | 88.77 | 87.17 | **93.05** |
| EDGE | 72.83 | 73.00 | 73.00 | 95.33 | 93.83 | 98.67 | 66.31 | 21.93 | 71.89 | 88.24 | 86.10 | 93.05 |
| $\max_{\mathcal{A}_\mathcal{N}}$ | 92.50 | 87.00 | 91.17 | 95.83 | 97.67 | 98.83 | 83.42 | 57.75 | 82.95 | 88.77 | 91.98 | 96.26 |
| RS | 87.33 | 81.83 | 90.67 | 95.50 | 95.83 | **98.83** | 75.40 | 43.85 | **78.34** | 88.77 | 87.70 | **95.19** |
| EDGE | 92.00 | 84.17 | 90.75 | 95.67 | 96.67 | 98.83 | 77.54 | 45.45 | 76.04 | 88.77 | 91.44 | 95.19 |
| $JSD_{RS}$ | 0.44 | 0.38 | 0.34 | **0.00** | 0.15 | **0.00** | 0.23 | 0.65 | 0.41 | **0.00** | 0.37 | 0.57 |
| $JSD_{EDGE}$ | **0.30** | **0.28** | **0.22** | **0.00** | 0.15 | **0.00** | **0.21** | **0.20** | **0.18** | **0.00** | **0.17** | **0.36** |
| $W_{RS}$ | 2.81 | 2.92 | 3.89 | **0.00** | 0.48 | **0.00** | 1.59 | 9.85 | 2.44 | **0.00** | 1.18 | 2.25 |
| $W_{EDGE}$ | **2.00** | **2.03** | **1.42** | **0.00** | **0.22** | **0.00** | **1.74** | **2.37** | **1.11** | **0.00** | **0.77** | **1.07** |

easiest sequence. To ensure the total number of sampled sequences remains unchanged, we randomly select one sequence as the *Median Sequence*, which is theoretically guaranteed to lie near the center of the distribution Theorem 1. By evaluating models on this triplet of task sequences, we approximate the true performance distribution and enable a more comprehensive assessment of model capability.

To generate hard sequences, we first cluster classes based on semantic similarity using hierarchical clustering (Nielsen, 2016). To encourage semantically similar classes to be grouped into the same task, we preserve large clusters intact and selectively split smaller ones as needed, ensuring all classes are assigned to $K$ tasks while minimizing global inter-task similarity.

After constructing task partitions, we compute the inter-task similarity matrix $\mathbf{ITS} \in \mathbb{R}^{K \times K}$ and initialize the sequence with the task exhibiting the lowest global similarity. Subsequent tasks are iteratively selected based on minimal similarity to the current task, forming candidate sequences.

By varying the clustering granularity, we generate multiple candidates and select the one with the lowest overall similarity score $\mathcal{S}(o)$. The easy sequence is constructed analogously, except that similar classes are intentionally assigned to different tasks, and each next task is selected based on maximal similarity to the previous one. *Pseudo-code and analysis are provided in the Appendix.*

## 5 EXPERIMENT

Due to the exponential growth in the number of possible class sequences in CIL scenarios (as shown in Lemma 1), obtaining the true performance distribution under standard experimental settings is infeasible. We therefore divide our experimental evaluation into two parts. First, we conduct **fully enumerable** experiments on subsets of standard datasets, enabling quantitative analysis to validate the effectiveness of the proposed EDGE protocol. Second, we perform analytical experiments under standard benchmark settings, visually demonstrating EDGE's strong capability in capturing extreme performance cases.

### 5.1 ENUMERABLE EXPERIMENTS

#### 5.1.1 EXPERIMENTAL SETUP

**Dataset and Metrics.** We conduct experiments on the CIFAR-100 and ImageNet-R (Krizhevsky, 2009) datasets. For each dataset, we select the first six classes and partition them into three tasks, generating 90 possible task permutations, which we consider the true distribution ($\mathcal{D}_{true}$). Next, we apply the RS evaluation protocol (using random seeds 0, 42, and 1993 (Lai et al., 2025; Li & Zhou, 2025; McDonnell et al., 2024; Wang et al., 2022e)) to generate class sequences for evaluation, obtaining the estimated distribution $\mathcal{D}_{RS}$. Simultaneously, we employ the EDGE protocol to perform th e evaluation, yielding the estimated distribution $\mathcal{D}_{EDGE}$. To quantitatively assess the effectiveness of different evaluation strategies, we use the JSD divergence and Wasserstein distance ($JSD_d$ (Lamberti et al., 2007) and $W_d$ (Villani, 2009)) to measure the differences between the estimated and true distributions.

Table 2: Experimental results of non-pre-trained-based methods on two datasets. Details are consistent with those in Table 1.

| Metric | CIFAR-100 | | | | | ImageNet-R | | | | | |
|---|---|---|---|---|---|---|---|---|---|---|---|
| | EWC | DER | iCaRL | FOSTER | MEMO | EWC | DER | iCaRL | FOSTER | MEMO | TagFex |
| $\min_{\mathcal{A_N}}$ | 12.50 | 16.83 | 36.33 | 16.00 | 21.83 | 10.06 | 11.73 | 10.61 | 11.17 | 12.85 | 11.73 |
| RS | 26.17 | **24.17** | 43.00 | 20.67 | 36.50 | 16.76 | **11.73** | **14.53** | 15.08 | **15.05** | 18.99 |
| EDGE | **12.50** | 26.35 | **38.50** | **16.67** | 35.67 | **10.61** | 11.97 | **14.53** | **11.73** | 15.44 | **14.53** |
| $\max_{\mathcal{A_N}}$ | 39.00 | 45.50 | 53.33 | 38.33 | 56.67 | 27.93 | 26.26 | 43.58 | 26.82 | 30.73 | 21.23 |
| RS | 27.50 | 34.17 | 43.00 | 23.50 | 51.17 | 21.23 | 18.99 | 22.91 | **24.02** | 21.23 | 20.11 |
| EDGE | **28.17** | **41.33** | **43.33** | **30.17** | **56.67** | **24.58** | **21.23** | **26.82** | 23.95 | **28.49** | **20.67** |
| $JSD_{RS}$ | 0.51 | **0.29** | 0.36 | 0.58 | 0.29 | 0.36 | 0.32 | 0.30 | 0.22 | 0.38 | 0.44 |
| $JSD_{EDGE}$ | **0.29** | 0.31 | **0.32** | **0.40** | **0.23** | **0.26** | **0.20** | **0.21** | **0.20** | **0.16** | **0.15** |
| $W_{RS}$ | 4.74 | 4.62 | 2.37 | 6.25 | 2.00 | 2.40 | 2.14 | 3.14 | 2.11 | 2.99 | 3.14 |
| $W_{EDGE}$ | **3.44** | **3.22** | **2.03** | **3.91** | **1.82** | **2.03** | **1.07** | **2.71** | **1.03** | **1.66** | **0.88** |

**Baseline.** To ensure a fair comparison, we benchmark our method under both non-pre-trained and pre-trained settings against classic and state-of-the-art approaches: in the non-pre-trained setting, we compare with EWC (Kirkpatrick et al., 2017), DER (Yan et al., 2021), iCaRL (Rebuffi et al., 2017), FOSTER (Wang et al., 2022a), MEMO (Zhou et al., 2023), and TagFex (Zheng et al., 2025); in the pre-trained setting, following Sun et al. (Sun et al., 2025a), we evaluate against L2P (Wang et al., 2022e), CODA-Prompt (Smith et al., 2023), HidePrompt (Wang et al., 2023b), EASE (Zhou et al., 2024c), and MOS (Sun et al., 2025b).

**Implementation Details.** Our framework is implemented in PyTorch, and the code is provided in the *supplementary materials*. Complete experimental details can be found in the *appendix*.

### 5.1.2 EXPERIMENT RESULTS

Tables 1 and 2 present the experimental results of two types of methods on the CIFAR-100 and ImageNet-R datasets. The results are organized into four sections: the first section shows the true lower performance bound (highlighted in gray) along with the lower bounds estimated by RS and EDGE; the second section similarly compares the upper performance bounds. The third and fourth sections display the JSD Divergence and the Wasserstein Distance, respectively, between the estimated distributions from both methods and the true distribution. Based on these experimental results, we draw the following conclusions:

- **RS leads to inaccurate and unfair comparisons.** RS produces biased estimates of performance boundaries, which may lead to unfair comparisons among models. For example, when evaluating non-pre-trained methods on the CIFAR-100 dataset, the lower bound estimated by RS for EWC (26.17%) is significantly higher than its true lower bound (12.50%). Notably, although the true lower bound of DER (16.83%) is actually better than that of EWC, the RS estimate suggests a worse lower bound for DER (24.17%), leading to an erroneous conclusion. In contrast, EDGE provides more accurate estimations of these boundaries, thereby avoiding such incorrect comparisons.

- **EDGE captures extremes and supports more comprehensive evaluation.** In the vast majority of experimental cases, the performance bounds estimated by EDGE are significantly closer to the ground-truth bounds (gray area) than those estimated by RS. Furthermore, EDGE demonstrates a stronger capability in approximating the true performance distribution, as reflected in its consistently lower or equal JSD Divergence and Wasserstein Distance values compared to RS in most scenarios.

- **Multiple methods may converge to similar worst-case performance under hard sequences.** On the challenging ImageNet-R dataset, the true lower-bound performance (i.e., worst-case accuracy) of multiple non-pre-trained methods clusters within a narrow range of 10.06% to 12.85%. This consistency suggests that task difficulty itself, rather than architectural differences, constitutes the primary bottleneck in this setting. EDGE helps model developers recognize this phenomenon, highlighting that variations in model design have limited impact under such conditions.

- **The accuracy of boundary estimation is correlated with model performance stability.** A model's performance stability directly affects how accurately its bounds can be estimated. Models with stable performance and low variance (e.g., EASE, MOS, and RanPAC in Table 1) enable both RS and EDGE to estimate bounds accurately, yielding near-zero JSD and Wasserstein Distance. In contrast, models with high performance fluctuation (e.g., non-pre-trained methods in Table 2) pose

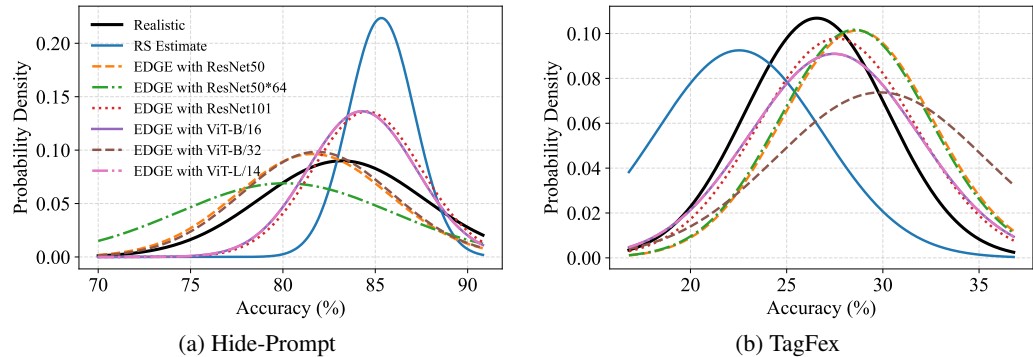

(a) Hide-Prompt

(b) TagFex

Figure 4: Effect of task sequences generated with CLIP text encoders of varying scales on the estimation of performance distributions under the EDGE protocol. The black curve denotes the ground-truth distribution, and the blue curve indicates the estimation obtained via the RS protocol.

greater challenges for bound estimation. It is in these cases that EDGE shows a clearer advantage over RS, producing closer bound estimates and lower distribution distances.

**EDGE is robust across different configurations.** Figure 4 demonstrates that the EDGE protocol maintains high estimation accuracy under various settings, including different model backbones (e.g., ResNet vs. ViT) and different sizes of the CLIP text encoder. In all cases, EDGE consistently outperforms the RS protocol by providing estimates that more closely align with the ground-truth performance distribution. This highlights the reliability and generalizability of EDGE across diverse model architectures and embedding capacities. *Additional results and detailed analyses for other experimental settings are provided in the appendix.*

## 5.2 EXPERIMENTS ON CLASSIC CIL SETTINGS

Following the classic CIL setup, we conduct experiments using three datasets: CIFAR-100 (Krizhevsky, 2009), CUB-200 (Wah et al., 2011), ImageNet-R (Krizhevsky, 2009). Each dataset is partitioned into multiple tasks of equal size. Figure 5 visualizes the maximum and minimum accuracy values ($\max_{\mathcal{A}}$ and $\min_{\mathcal{A}}$) of the sampled sequences under each protocol, highlighting their ability to capture the extremes of the performance distribution.The results demonstrate that EDGE consistently achieves both a lower estimated lower bound and a higher upper bound across nearly all scenarios, including highly stable methods such as EASE (Zhou et al., 2024c) and RanPAC (McDonnell et al., 2024). This allows it to identify rare but critical performance extremes, providing a more reliable and practical assessment of performance for real-world deployments. *For more detailed analysis, please refer to the Appendix.*

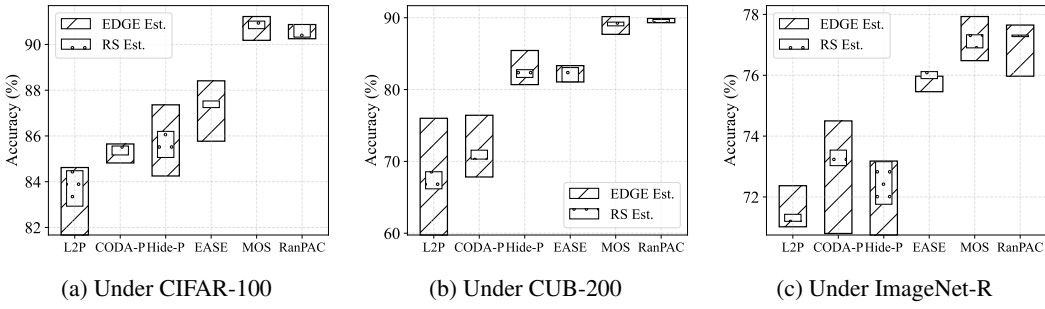

(a) Under CIFAR-100

(b) Under CUB-200

(c) Under ImageNet-R

Figure 5: Visualization of the estimated lower and upper performance bounds across three datasets under the classic CIL setting: (a) CIFAR-100, (b) CUB-200, and (c) ImageNet-R. The slashed bars (/) denote the proposed **EDGE**, while the dotted bars (.) correspond to the existing **RS** protocol.

# 6 CONCLUSIONS

Class incremental learning (CIL) is inherently sensitive to the class-arrival order, making evaluation a distributional problem rather than a single-point estimate. In this paper, we revisit the mainstream random sampling (RS) protocol and show, both theoretically and empirically, that estimating performance from only a few randomly drawn sequences can yield biased means, severely underestimated variance, and misleading conclusions about robustness under rare but critical class orders.

To address this issue, we propose EDGE, an extreme case–aware evaluation protocol that leverages inter-task similarity to construct representative easy, medium, and hard sequences, thereby providing a more faithful approximation of the underlying performance distribution and its boundaries. Our analysis establishes the key role of extreme sequences in sample-efficient distribution estimation, and motivates similarity-guided sequence construction. Extensive experiments on both fully enumerable settings and classic CIL benchmarks demonstrate that EDGE more reliably captures performance extremes and yields distribution estimates closer to the ground truth than RS, offering more actionable evidence for model comparison, selection, and robustness checking in realistic deployments.

## ACKNOWLEDGMENTS

This work is partially supported by NSFC (62376118, 62506160, 62476228), the Basic Research Program of Jiangsu (BK 20251251), and JSTJ-2025-147.

## ETHICS STATEMENT

This work does not involve human subjects, personally identifiable information, or sensitive data. All experiments were conducted on publicly available benchmark datasets under their respective licenses, and no private or restricted data were used. The proposed evaluation protocol, EDGE, is designed to provide more reliable assessments of continual learning models and does not directly introduce risks of harm. The authors affirm that this research complies with the ICLR Code of Ethics and with standard practices of research integrity and transparency.

## REPRODUCIBILITY STATEMENT

We are committed to ensuring the reproducibility of our work. To this end, we provide the following: (1) **Code availability**: Our code is publicly available at `https://github.com/AIGNLAI/EDGE`. Detailed instructions for reproducing our experiments are provided in Section E. (2) **Theoretical results**: All assumptions are clearly stated, and complete proofs of our main theorems are presented in Sections B and C.1. (3) **Experimental details and additional results**: Further dataset descriptions, additional experiments, and analyses are provided in Sections B.2 and D.1.2.

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

APPENDIX

The appendix is organized as follows:

- Section A introduces the notations and mathematical symbols used throughout the paper, providing a clear reference for theoretical and algorithmic components.
- Section B presents a detailed analysis of the existing RS (Random Sampling) protocol, including formal proofs of its limitations and additional empirical results that support our claims.
- Section C provides a comprehensive discussion of the proposed EDGE protocol. This includes pseudo-code, step-by-step explanations of the sequence generation algorithm, and theoretical justification for its effectiveness.
- Section D reports extended experimental results and offers in-depth analysis of the observed patterns. These findings provide new insights into the design and selection of CIL algorithms under varying sequence difficulties.
- Section E provides a practical guide to using our open-source EDGE repository, including installation, running commands for the random vs. EDGE protocols, and notes on the currently supported datasets and how to extend EDGE to new datasets.

## A  NOTATION

Table A1: Notations and their explanations used throughout this paper.

| Notation | Explanation |
|---|---|
| $N$ | Total number of classes. |
| $\mathcal{N}(\mu, \sigma^2)$ | Gaussian (normal) distribution with mean $\mu$ and variance $\sigma^2$. |
| $\mathcal{D}^i$ | Training dataset for task $i$, consisting of $n_i$ input–label pairs. |
| $CLS^i$ | Set of classes associated with task $i$. |
| $\Omega$ | Sample space of all possible class sequences. |
| $\mathcal{O} \in \Omega$ | A specific ordered class sequence of length $K$. |
| $\mathcal{P}_{\text{true}}$ | True (but unknown) distribution of the model's performance. |
| $\mathcal{A}(\mathcal{O})$ | Final accuracy achieved by the model on sequence $\mathcal{O}$. |
| $K$ | Total number of tasks (i.e., the length of each sequence). |
| $L$ | Number of sampled sequences used for estimation. |
| $\delta$ | Allowed failure probability (so confidence is $1 - \delta$). |
| $\epsilon$ | Tolerance for estimation error. |
| $\mathcal{S}(\mathcal{O})$ | Inter-task similarity score for sequence $\mathcal{O}$. |
| $\epsilon_g$ | Theoretical upper bound on the generalization error. |
| $\Phi$ | CLIP text encoder mapping text tokens to $d$-dimensional embeddings. |
| $\mathbf{L}$ | Matrix of text embeddings for the $C$ class labels. |
| $\mathbf{D}$ | Similarity matrix between label embeddings (e.g. cosine similarity). |
| $\mathbf{ITS}$ | Inter-task similarity matrix aggregated from $\mathbf{D}$. |

Table A1 provides a detailed description of the notations used throughout the paper, facilitating a clearer understanding of the mathematical formulations and algorithmic procedures.

## B  DETAILED ANALYSIS OF RS PROTOCOL

### B.1  THEORETICAL ANALYSIS AND PROOF

**Lemma 1.** *Let $N$ be the total number of classes, partitioned into $K$ tasks of equal size $M = N/K$. Then the number of distinct class sequences is $|\Omega| = \frac{N!}{(M!)^K}$. Moreover, under linear scaling*

$K = \Theta(N)$, *the quantity* $|\Omega|$ *grows factorially, satisfying* $|\Omega| = \Omega\big((N/e)^N\big)$, *which asymptotically dwarfs any polynomial-scale sampling capacity as* $N \to \infty$.

*Proof.* We begin by noting that the total number of distinct permutations of $N$ classes is $N!$. When partitioning these classes into $K$ tasks of equal size $M = N/K$, each individual task contains $M$ unordered classes. Since the order of classes within a task is irrelevant but the order of tasks themselves is preserved, we must quotient out the intra-task permutations.

For each task, there are $M!$ ways to permute its classes internally. Since there are $K$ such tasks, the total number of intra-task permutations is $(M!)^K$. Consequently, the total number of distinct class sequences that respect this task-based structure is given by $|\Omega| = \frac{N!}{(M!)^K}$. To analyze the growth rate of $|\Omega|$, assume a linear scaling regime where $K = \Theta(N)$. Then $M = N/K = \Theta(1)$, implying that $M!$ is constant with respect to $N$. Therefore, $(M!)^K = \Theta(c^N)$ for some constant $c > 0$.

By Stirling's approximation, we have:

$$N! \sim \sqrt{2\pi N}\left(\frac{N}{e}\right)^N. \tag{6}$$

Thus,

$$|\Omega| = \frac{N!}{(M!)^K} = \Omega\left(\frac{(N/e)^N}{c^N}\right) = \Omega\left(\left(\frac{N}{ec}\right)^N\right), \tag{7}$$

which shows that $|\Omega|$ grows at least as fast as $(N/e')^N$ for some constant $e' > e$, i.e., $|\Omega| = \Omega\left((N/e)^N\right)$.

Finally, note that any polynomial function in $N$ is dominated by $(N/e)^N$ as $N \to \infty$. Hence, the number of possible class sequences $|\Omega|$ asymptotically exceeds any polynomial-scale sampling budget, concluding the proof. $\square$

**Theorem 1.** *Let* $\Omega$ *be the set of all possible class sequences with* $|\Omega|$ *elements, and fix tolerance* $\varepsilon > 0$ *and failure probability* $\delta \in (0,1)$. *Suppose we draw* $L$ *sequences* $\{RS_l\}_{l=1}^L$ *without* replacement *uniformly from* $\Omega$, *and let* $\widehat{\mathcal{A}}_L = \frac{1}{L}\sum_{l=1}^L \mathcal{A}(RS_l)$ *be the empirical mean, respectively. Then for any* $\varepsilon > 0$, *if*

$$L\,\frac{|\Omega| - L}{|\Omega| - 1} \geq \frac{\ln\big(2\,|\Omega|/\delta\big)}{2\,\varepsilon^2}, \tag{8}$$

*then with probability at least* $1 - \delta$, $\big|\widehat{\mathcal{A}}_L - \mathbb{E}_\omega[\mathcal{A}(\omega)]\big| \leq \varepsilon$.

*Proof.* Let $\Omega = \{\omega_1, \omega_2, \ldots, \omega_{|\Omega|}\}$ denote the finite set of all possible class sequences. Define the true mean accuracy as

$$\mu = \mathbb{E}_{\omega \sim \Omega}[\mathcal{A}(\omega)] = \frac{1}{|\Omega|}\sum_{i=1}^{|\Omega|} \mathcal{A}(\omega_i). \tag{9}$$

Let $\{RS_1, RS_2, \ldots, RS_L\}$ be a sample of size $L$ drawn uniformly at random *without replacement* from $\Omega$, and define the empirical mean

$$\widehat{\mathcal{A}}_L = \frac{1}{L}\sum_{i=1}^L \mathcal{A}(RS_i). \tag{10}$$

Our goal is to bound the deviation probability $\mathbb{P}(|\widehat{\mathcal{A}}_L - \mu| \geq \varepsilon)$, under the assumption that $\mathcal{A}(\cdot) \in [0,1]$ for all $\omega \in \Omega$. Let us define the Doob martingale sequence

$$Z_0 = \mathbb{E}[\widehat{\mathcal{A}}_L], \quad Z_i = \mathbb{E}[\widehat{\mathcal{A}}_L \mid RS_1, \ldots, RS_i], \quad i = 1, \ldots, L. \tag{11}$$

Then $\{Z_i\}_{i=0}^L$ forms a martingale with respect to the filtration $\mathcal{F}_i = \sigma(RS_1, \ldots, RS_i)$, and we have:

$$Z_0 = \mu, \quad Z_L = \widehat{\mathcal{A}}_L, \quad \text{and} \quad \widehat{\mathcal{A}}_L - \mu = Z_L - Z_0 = \sum_{i=1}^L (Z_i - Z_{i-1}). \tag{12}$$

Since the sampling is without replacement from a bounded set $\mathcal{A}(\omega) \in [0, 1]$, we can bound each martingale difference:

$$|Z_i - Z_{i-1}| \leq \frac{1}{L} \cdot \sqrt{\frac{|\Omega| - L}{|\Omega| - 1}}, \quad \text{for all } i = 1, \ldots, L. \tag{13}$$

This bound can be obtained via an extension of McDiarmid's inequality for sampling without replacement, or directly computed via sensitivity analysis of the sample mean with respect to one replacement in the sequence. Thus, the variance proxy is bounded as:

$$\sum_{i=1}^{L} (Z_i - Z_{i-1})^2 \leq L \cdot \left( \frac{1}{L^2} \cdot \frac{|\Omega| - L}{|\Omega| - 1} \right) = \frac{1}{L} \cdot \frac{|\Omega| - L}{|\Omega| - 1}. \tag{14}$$

Using the standard Azuma–Hoeffding inequality for martingales with bounded increments, we obtain:

$$\mathbb{P}\left( \left| \widehat{\mathcal{A}}_L - \mu \right| \geq \varepsilon \right) \leq 2 \exp\left( -\frac{\varepsilon^2}{2 \sum_{i=1}^{L} (Z_i - Z_{i-1})^2} \right) \leq 2 \exp\left( -2\varepsilon^2 L \cdot \frac{|\Omega| - L}{|\Omega| - 1} \right). \tag{15}$$

To ensure that the deviation probability is at most $\delta/|\Omega|$ for each of the $|\Omega|$ possible values (for use in a union bound), it suffices that:

$$2 \exp\left( -2\varepsilon^2 L \cdot \frac{|\Omega| - L}{|\Omega| - 1} \right) \leq \frac{\delta}{|\Omega|}. \tag{16}$$

Solving this inequality, we take logarithms on both sides:

$$-2\varepsilon^2 L \cdot \frac{|\Omega| - L}{|\Omega| - 1} \leq \ln(\delta/2S), \tag{17}$$

which is equivalent to:

$$L \cdot \frac{|\Omega| - L}{|\Omega| - 1} \geq \frac{\ln(2|\Omega|/\delta)}{2\varepsilon^2}. \tag{18}$$

$\square$

**Theorem 2.** *Let $\Omega$ be the set of all class sequences, and define $\mu = \mathbb{E}_{\omega \sim \Omega}[\mathcal{A}(\omega)]$, $\sigma = \sqrt{\mathrm{Var}_{\omega \sim \Omega}[\mathcal{A}(\omega)]}$ as the realistic mean and standard deviation of the accuracy function $\mathcal{A}$. Suppose we know two extreme sequences $\omega_+$, $\omega_-$ satisfying $\mathcal{A}(\omega_+) - \mu \geq \sigma$ and $\mu - \mathcal{A}(\omega_-) \geq \sigma$. Draw $L$ sequences $\{RS_l\}_{l=1}^{L}$ without replacement uniformly from $\Omega \setminus \{\omega_+, \omega_-\}$, and define $\widetilde{\mathcal{A}}_{L+2} = \frac{1}{L+2}\left[ \mathcal{A}(\omega_-) + \mathcal{A}(\omega_+) + \sum_{l=1}^{L} \mathcal{A}(RS_l) \right]$. Then for any $\varepsilon > 0$ and $\delta \in (0, 1)$, if*

$$L \frac{|\Omega| - 2 - L}{|\Omega| - 3} \geq \frac{\ln\left( 2\left(|\Omega| - 2\right)/\delta \right) \left( R^{(\sigma)} \right)^2}{2\varepsilon^2}, \tag{19}$$

*where $R^{(\sigma)} = \mathcal{A}(\omega_+) - \mathcal{A}(\omega_-)$, then with probability at least $1 - \delta$, $\left| \widetilde{\mathcal{A}}_{L+2} - \mathbb{E}_{\omega \sim \Omega}[\mathcal{A}(\omega)] \right| \leq \varepsilon$.*

*Proof.* Let $\Omega = \{\omega_1, \ldots, \omega_{|\Omega|}\}$ and write

$$\mu = \frac{1}{|\Omega|} \sum_{i=1}^{|\Omega|} \mathcal{A}(\omega_i), \quad \sigma^2 = \mathrm{Var}_{\omega \sim \Omega}[\mathcal{A}(\omega)] \tag{20}$$

By assumption there exist sequences $\omega_+, \omega_-$ satisfying

$$\mathcal{A}(\omega_+) - \mu \geq \sigma, \quad \mu - \mathcal{A}(\omega_-) \geq \sigma. \tag{21}$$

Define the total range

$$R^{(\sigma)} = \mathcal{A}(\omega_+) - \mathcal{A}(\omega_-) \geq 2\sigma. \tag{22}$$

Draw $L$ samples $\{RS_i\}_{i=1}^{L}$ without replacement from $\Omega' = \Omega \setminus \{\omega_+, \omega_-\}$, and set

$$\widetilde{\mathcal{A}}_{L+2} = \frac{1}{L+2}\left[ \mathcal{A}(\omega_-) + \mathcal{A}(\omega_+) + \sum_{i=1}^{L} \mathcal{A}(RS_i) \right]. \tag{23}$$

Consider the Doob martingale

$$Z_0 = \mathbb{E}[\widetilde{\mathcal{A}}_{L+2}], \quad Z_i = \mathbb{E}[\widetilde{\mathcal{A}}_{L+2} \mid RS_1, \ldots, RS_i], \quad i = 1, \ldots, L. \tag{24}$$

Then

$$Z_0 = \mu, \quad Z_L = \widetilde{\mathcal{A}}_{L+2}, \quad \widetilde{\mathcal{A}}_{L+2} - \mu = \sum_{i=1}^{L}(Z_i - Z_{i-1}). \tag{25}$$

*Since each $\mathcal{A}(RS_i)$ lies between the two extremes, replacing one sample can change the sum by at most $R^{(\sigma)}$. Moreover, because we sample without replacement from a set of size $|\Omega| - 2$, the sensitivity of the average $\widetilde{\mathcal{A}}_{L+2}$ to a single replacement is further scaled by $\sqrt{(|\Omega| - 2 - L)/(|\Omega| - 3)}$. Altogether one obtains*

$$|Z_i - Z_{i-1}| \leq \frac{1}{L+2} R^{(\sigma)} \sqrt{\frac{|\Omega| - 2 - L}{|\Omega| - 3}}, \quad i = 1, \ldots, L. \tag{26}$$

Hence, the sum of squared increments is bounded by

$$\sum_{i=1}^{L}(Z_i - Z_{i-1})^2 \leq L \left(\frac{R^{(\sigma)}}{L+2}\right)^2 \frac{|\Omega| - 2 - L}{|\Omega| - 3}. \tag{27}$$

By Azuma–Hoeffding,

$$\Pr\left(|\widetilde{\mathcal{A}}_{L+2} - \mu| \geq \varepsilon\right) \leq 2\exp\left(-\frac{\varepsilon^2}{2\sum_{i=1}^{L}(Z_i - Z_{i-1})^2}\right) \leq 2\exp\left(-2\varepsilon^2 L \frac{|\Omega| - 2 - L}{|\Omega| - 3} (R^{(\sigma)})^{-2}\right). \tag{28}$$

Requiring this probability to be at most $\delta/(|\Omega| - 2)$ and solving for $L$ gives

$$L \frac{|\Omega| - 2 - L}{|\Omega| - 3} \geq \frac{\ln(2(|\Omega| - 2)/\delta)(R^{(\sigma)})^2}{2\varepsilon^2}. \tag{29}$$

$\square$

## B.2 EMPIRICAL ANALYSIS

To empirically validate the distributional characteristics of performance metrics across different continual learning methods, we conducted experiments on CIFAR-100 and ImageNet-R. As shown in Table A2, the number of possible class sequences grows rapidly even with a small number of classes. To balance feasibility and distributional richness, we chose a configuration with 6 classes divided into 3 tasks of 2 classes each, yielding 90 possible class sequences. This setting is large enough to exhibit meaningful variation in performance, yet still allows complete enumeration of the sequence space. For each dataset, we randomly selected 6 classes and partitioned them accordingly. We then evaluated two groups of methods:

Table A2: Number of possible class sequences $|\Omega|$ under different partitions

| $N$ | $K$ | $M = N/K$ | Formula | $|\Omega| = \frac{N!}{(M!)^K}$ |
|-----|-----|-----------|---------|--------------------------------|
| 4 | 2 | 2 | $\frac{4!}{(2!)^2}$ | 6 |
| 6 | 2 | 3 | $\frac{6!}{(3!)^2}$ | 20 |
| 8 | 2 | 4 | $\frac{8!}{(4!)^2}$ | 70 |
| 10 | 2 | 5 | $\frac{10!}{(5!)^2}$ | 252 |
| **6** | **3** | **2** | $\frac{6!}{(2!)^3}$ | **90** |
| 9 | 3 | 3 | $\frac{9!}{(3!)^3}$ | 1680 |
| 8 | 4 | 2 | $\frac{8!}{(2!)^4}$ | 2520 |

- **Non-pretrained CIL methods**: EWC Kirkpatrick et al. (2017), DER Yan et al. (2021), iCaRL Rebuffi et al. (2017), FOSTER Wang et al. (2022a), MEMO Zhou et al. (2023), and TagFex Zheng et al. (2025).

- **Pre-trained CIL methods** : L2P Wang et al. (2022e), CODA-Prompt Smith et al. (2023), Hide-Prompt Wang et al. (2023b), EASE Zhou et al. (2024c), and MOS Sun et al. (2025b).

**Observation 1: The capacity distribution of the model is near-Gaussian**

For each method–dataset pair, we collected the final task accuracies over the 90 sequences and applied a Box–Cox power transformation. The optimal parameter $\lambda$ was chosen by maximizing the log-likelihood under the normality assumption. We then performed three normality tests on the transformed accuracies: the Shapiro–Wilk test, D'Agostino's $K^2$ test, and the one-sample Kolmogorov–Smirnov (KS) test. Each yields a $p$-value indicating the probability of observing the data under a Gaussian null hypothesis.

Table A3: Normality test results, **demonstrating that the model capacity distribution approximates a Gaussian**

| Method | Dataset | $\lambda$ | Shapiro–Wilk $p$ | D'Agostino's $p$ | KS $p$ |
|---|---|---|---|---|---|
| CODA-Prompt | CIFAR-100 | 4.5971 | 0.4753 | 0.4357 | 0.8365 |
| CODA-Prompt | ImageNet-R | 2.3957 | 0.9321 | 0.7483 | 0.8811 |
| DER | CIFAR-100 | 2.2577 | 0.3582 | 0.3030 | 0.3824 |
| DER | ImageNet-R | 0.0321 | 0.2755 | 0.5488 | 0.6937 |
| EWC | CIFAR-100 | 1.5076 | 0.1926 | 0.1913 | 0.4064 |
| EWC | ImageNet-R | 0.7893 | 0.2854 | 0.5735 | 0.3519 |
| FOSTER | CIFAR-100 | 2.8988 | 0.2215 | 0.9340 | 0.4010 |
| FOSTER | ImageNet-R | 0.1042 | 0.2216 | 0.1903 | 0.2880 |
| Hide-Prompt | CIFAR-100 | 4.1164 | 0.5802 | 0.4904 | 0.4783 |
| Hide-Prompt | ImageNet-R | 1.7462 | 0.3538 | 0.8186 | 0.6373 |
| iCaRL | CIFAR-100 | 7.2381 | 0.1369 | 0.0326 | 0.4413 |
| iCaRL | ImageNet-R | 1.1917 | 0.9577 | 0.9403 | 0.9870 |
| L2P | CIFAR-100 | 11.8644 | 0.0554 | 0.3215 | 0.2628 |
| L2P | ImageNet-R | 4.4415 | 0.1850 | 0.1541 | 0.6715 |
| MEMO | CIFAR-100 | 0.7041 | 0.1720 | 0.0524 | 0.7084 |
| MEMO | ImageNet-R | 0.6267 | 0.8781 | 0.8697 | 0.7357 |
| MOS | CIFAR-100 | -12.9763 | 0.4968 | 0.7681 | 0.5230 |
| MOS | ImageNet-R | 3.0909 | 0.1546 | 0.7973 | 0.2497 |

The results in Table A3 indicate that, after Box–Cox transformation, most method–dataset combinations exhibit $p$-values above the conventional 0.05 threshold in at least two of the three tests, suggesting an adequate approximation to normality.

In addition, methods not listed in Table A3, such as RanPAC and EASE, produce a limited number of possible task sequences due to their architectural design, resulting in insufficient sample sizes for reliable normality testing; hence, their results are reported as *n/a*.

**Observation 2: The sampling of RS protocol cannot reflect the realistic ability of the model well**

Figure A1 and Figure A2 illustrate the true performance distribution and the sampling locations obtained using the RS protocol (random seeds 0, 42, and 1993). Random sampling often fails to capture the true characteristics of the distribution. First, most sampled points cluster around the center of the distribution, making them ineffective in reflecting the model's behavior under extreme conditions. Second, the randomness of the sampling process introduces significant uncertainty across different data types and datasets, as the sampling locations vary considerably. This variability leads to unstable evaluations, where some methods are overestimated while others are underestimated. Third, two major issues arise when using these randomly selected points to estimate the true distribution: the mean is inaccurately estimated, and the variance is severely underestimated. These problems together compromise the reliability of model evaluation under the RS protocol.

**Observation 3: The performance of the model is positively correlated with inter-task similarity**

Building upon the prior theoretical analysis and the first two observations, we recognize the necessity of incorporating extreme sequences for auxiliary evaluation. However, a key challenge lies in adaptively identifying such extreme sequences and determining a principled basis for algorithmic design. In the CIL setting, it is intuitively understood that when adjacent tasks exhibit low similarity, the model parameters undergo substantial changes during task transitions, increasing the risk of

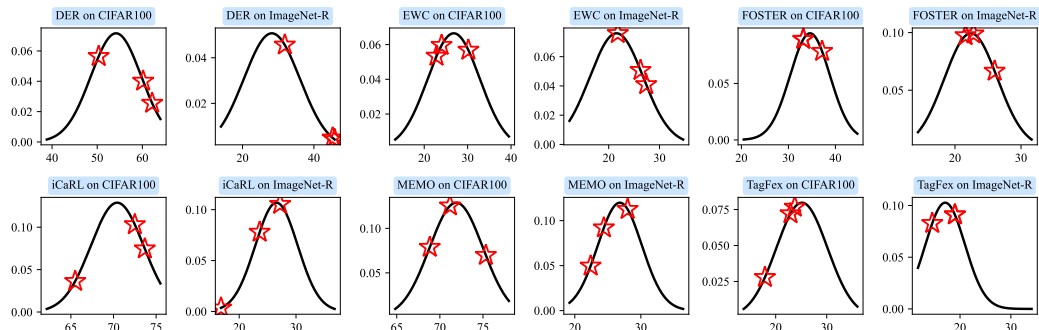

Figure A1: True performance distribution (black) and sampling positions under the RS protocol (blue) for non-pretrained CIL methods. The figure illustrates that RS fails to adequately capture the true distribution, leading to biased estimation.

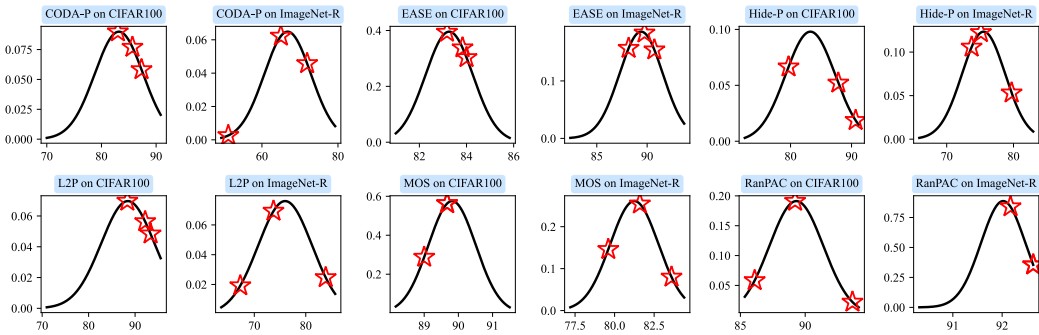

Figure A2: True data distribution and the sampling positions under the RS protocol (pre-trained CIL methods).

forgetting. Motivated by this intuition, we investigate the relationship between inter-task similarity and model performance. As shown in Figure 2c, a strong positive correlation is observed across most methods. This insight suggests that inter-task similarity can be a foundation for designing strategies to sample challenging sequences and evaluate model robustness.

## C DETAILED ANALYSIS OF EDGE

### C.1 THEORETICAL ANALYSIS AND PROOF

**Theorem 3.** *Consider a CIL system consisting of $K$ tasks, where each task $T_k$ is associated with a data distribution $\mathcal{D}_k$ and a class set $\mathcal{C}_k$. The generalization error is defined as $\epsilon_g = \frac{1}{K}\sum_{k=1}^{K}\mathbb{E}_{(x,y)\sim\mathcal{D}_k}[L(h(x),y)]$, where $L(h(x),y)$ denotes the loss between the model prediction $h(x)$ and the true label $y$. Given a task order $\mathcal{O} = \{T_1, T_2, \ldots, T_K\}$, the similarity score $\mathcal{S}(\mathcal{O})$ is defined as:*

$$\mathcal{S}(\mathcal{O}) = \frac{K}{(K-1)N} \sum_{1\leq i\leq K-1} \sum_{c\in\mathcal{C}_i} \sum_{c'\in\mathcal{C}_{i+1}} Sim(c,c'), \tag{30}$$

*where $Sim(c,c')$ denotes the semantic similarity in the representation space between classes $c$ and $c'$, belonging to tasks $T_i$ and $T_j$, respectively. Let $\mathcal{O}_h$ and $\mathcal{O}_e$ denote the sequences with the minimum and maximum similarity scores $\mathcal{S}(\mathcal{O})$, respectively, and let $\mathcal{O}_r$ represent a randomly generated sequence. Then, the following conditions hold:*

- *The similarity score satisfies $\mathcal{S}(\mathcal{O}_h) \leq \mathcal{S}(\mathcal{O}_r) \leq \mathcal{S}(\mathcal{O}_e)$,*

- *The generalization error satisfy $\epsilon_g(\mathcal{O}_h) \geq \epsilon_g(\mathcal{O}_r) \geq \epsilon_g(\mathcal{O}_e)$.*

**Lemma 2.** *Lin et al. (2023) When $p \geq n + 2$, we must have:*

$$\mathbb{E}[\epsilon_g] = \frac{r^T}{T} \sum_{i=1}^{T-1} \|w_i^*\|^2 + \frac{1-r}{T} \sum_{i=1}^{T} r^{T-i} \sum_{k=1}^{T} \|w_k^* - w_i^*\|^2$$

$$+ \frac{p\sigma^2}{p-n-1}(1 - r^T). \tag{31}$$

*where the overparameterization ratio $r = 1 - \frac{n}{p}$ in this context quantifies the degree of overparameterization in a model, where $n$ represents the sample size, and $p$ denotes the number of model parameters. The coefficients $c_{i,j} = (1-r)(r^{T-i} - r^{j-i} + r^{T-j})$, with $1 \leq i < j \leq T$, correspond to the indices of tasks, and $\sigma$ denotes a coefficient representing the model's noise level.*

*Proof.* First, consider the total cross-task similarity:

$$\sum_{1 \leq i \leq K} \sum_{1 \leq j \leq K} \sum_{c \in \mathcal{C}_i} \sum_{c' \in \mathcal{C}_j} Sim(c, c'), \tag{32}$$

and the intra-task similarity:

$$\sum_{1 \leq i \leq K} \sum_{c,c' \in \mathcal{C}_i} Sim(c, c'). \tag{33}$$

These satisfy the conservation relationship:

$$\sum_{1 \leq i \leq K} \sum_{1 \leq j \leq K} \sum_{c \in \mathcal{C}_i} \sum_{c' \in \mathcal{C}_j} Sim(c, c') + \sum_{1 \leq i \leq K} \sum_{c,c' \in \mathcal{C}_i} Sim(c, c') = C_1. \tag{34}$$

For optimal parameters between tasks $i$ and $j$, we have:

$$\|w_i^* - w_j^*\|^2 \propto \|\sum_{m \in \mathcal{C}_i} v_m^* - \sum_{n \in \mathcal{C}_j} v_n^*\|^2$$

$$= \sum_{m \in \mathcal{C}_i} \sum_{n \in \mathcal{C}_i} \langle v_m, v_n \rangle + \sum_{m \in \mathcal{C}_j} \sum_{n \in \mathcal{C}_j} \langle v_m, v_n \rangle - 2 \sum_{m \in \mathcal{C}_i} \sum_{n \in \mathcal{C}_j} \langle v_m, v_n \rangle$$

$$= \alpha \left( \sum_{c,c' \in \mathcal{C}_i} Sim(c, c') + \sum_{c,c' \in \mathcal{C}_j} Sim(c, c') \right) - \alpha \sum_{c \in \mathcal{C}_i} \sum_{c' \in \mathcal{C}_j} Sim(c, c'), \tag{35}$$

where $\alpha$ is a proportionality constant.

Substituting into the key term from Lemma 2:

$$\frac{1-r}{K} \sum_{i=1}^{K} r^{K-i} \sum_{k=1}^{K} \|w_k^* - w_i^*\|^2$$

$$= \frac{1-r}{K} \sum_{i=1}^{K} r^{K-i} \sum_{k=1}^{K} \alpha \left( \sum_{c,c' \in \mathcal{C}_i} Sim(c, c') + \sum_{c,c' \in \mathcal{C}_k} Sim(c, c') - \sum_{c \in \mathcal{C}_i} \sum_{c' \in \mathcal{C}_k} Sim(c, c') \right)$$

$$= \frac{1-r}{K} \sum_{i=1}^{K} r^{K-i} \alpha \left( (K-1) \sum_{c,c' \in \mathcal{C}_i} Sim(c, c') - \sum_{k=1}^{K} \sum_{c \in \mathcal{C}_i} \sum_{c' \in \mathcal{C}_k} Sim(c, c') \right)$$

$$= \frac{1-r^K}{K} \alpha(K-1) \left( C_1 - \sum_{i=1}^{K} \sum_{j=1,j \neq i}^{K} \sum_{c \in \mathcal{C}_i} \sum_{c' \in \mathcal{C}_j} Sim(c, c') \right)$$

$$- \frac{1-r}{K} \sum_{i=1}^{K} r^{K-i} \alpha \sum_{k=1}^{K} \sum_{c \in \mathcal{C}_i} \sum_{c' \in \mathcal{C}_k} Sim(c, c')$$

$$= (1 - r^K)\alpha C_1 - \frac{1-r}{K} \alpha \sum_{i=1}^{K} \sum_{k=1,k \neq i}^{K} r^{k-i} \sum_{c \in \mathcal{C}_i} \sum_{c' \in \mathcal{C}_k} Sim(c, c')$$

$$= C_2 - 2\frac{r - r^2}{K}\alpha \sum_{1 \leq i \leq K-1} \sum_{c \in \mathcal{C}_i} \sum_{c' \in \mathcal{C}_{i+1}} Sim(c, c'), \tag{36}$$

where $C_2$ contains terms independent of the task ordering.

To establish the probabilistic bound for random sequences, let $\Omega$ denote the set of all possible task permutations and $\mathcal{O}_r \sim \text{Unif}(\Omega)$. Define the random variable $X_{i,j} = \sum_{c \in \mathcal{C}_i} \sum_{c' \in \mathcal{C}_j} Sim(c, c')$. The similarity score can be rewritten as:

$$\mathcal{S}(\mathcal{O}) = \frac{K}{(K - 1)N} \sum_{t=1}^{K-1} X_{\pi(t),\pi(t+1)}, \tag{37}$$

where $\pi$ is the permutation function. By symmetry, the probability that any two distinct tasks $T_i$ and $T_j$ are adjacent in a random permutation is $\frac{2}{K(K-1)}$. Thus, the expected similarity score is:

$$\mathbb{E}[\mathcal{S}(\mathcal{O}_r)] = \frac{K}{(K - 1)N} \cdot \frac{2}{K(K - 1)} \sum_{1 \leq i < j \leq K} X_{i,j}$$

$$= \frac{2}{(K - 1)^2 N} \sum_{i \neq j} X_{i,j}. \tag{38}$$

Applying McDiarmid's inequality to the function $f(\pi) = \mathcal{S}(\mathcal{O})$, observe that swapping two tasks in $\pi$ changes $f(\pi)$ by at most $\frac{4U}{N}$, where $U$ is the upper bound of $Sim(c, c')$. This yields:

$$\mathbb{P}\left(|f(\pi) - \mathbb{E}[f]| \geq \delta\right) \leq 2\exp\left(-\frac{2\delta^2 N^2}{K(4U)^2}\right). \tag{39}$$

Letting $\delta = \min(\mathbb{E}[f] - \mathcal{S}(\mathcal{O}_h), \mathcal{S}(\mathcal{O}_e) - \mathbb{E}[f])$, we obtain the concentration bound:

$$\mathbb{P}\left(\mathcal{S}(\mathcal{O}_h) \leq \mathcal{S}(\mathcal{O}_r) \leq \mathcal{S}(\mathcal{O}_e)\right) \geq 1 - 2\exp\left(-\frac{K\delta^2}{8U^2}\right). \tag{40}$$

This reveals an inverse relationship between the similarity score $\mathcal{S}(\mathcal{O})$ and the generalization error $\epsilon_g$: the coefficient before the similarity summation term is negative, meaning higher similarity scores correspond to lower generalization error. Therefore, the ordering with maximum similarity $\mathcal{O}_e$ minimizes $\epsilon_g$, while the minimum similarity ordering $\mathcal{O}_h$ maximizes $\epsilon_g$, with random ordering $\mathcal{O}_r$ falling between them. □

## C.2 PSEUDO CODE AND ANALYSIS

**Algorithm Analysis.** The proposed algorithm generates hard task sequences by systematically minimizing inter-task similarities, which aligns with Theorem 1's conclusion that lower similarity scores correspond to higher generalization error. Key design rationales are analyzed as follows:

- **Step 2-3 (Dissimilarity Computation):** First, we convert the class similarity matrix $\mathbf{G} \in \mathbb{R}^{N \times N}$ into a dissimilarity matrix by computing $\mathbf{D} = \mathbf{1} - \mathbf{G}$ and setting the diagonal to zero. This transformation ensures compatibility with clustering algorithms, where larger values indicate greater dissimilarity. Next, we apply hierarchical clustering with complete linkage on the condensed form of $\mathbf{D}$ to obtain $K$ clusters. These clusters are then sorted by size in descending order. For clusters whose size exceeds the base task size $M = N/K$, we iteratively assign subsets of the cluster to the currently smallest task to preserve internal semantic similarity while maintaining balance. For smaller clusters, we assign all classes directly to the current shortest task. After this initial allocation, we perform a final adjustment step to ensure all tasks are of equal size: any task exceeding the base size has its excess classes redistributed to tasks with fewer classes. This process results in $K$ balanced tasks, each composed of semantically coherent classes, effectively minimizing global inter-task similarity and supporting the construction of hard class sequences.
- **Step 4-5 (Multi-Granularity Clustering):** Varying granularity levels $g \in \mathcal{G}$ enable exploration of different class grouping resolutions. This multi-scale approach increases the probability of discovering optimal task boundaries that minimize cross-task similarities.

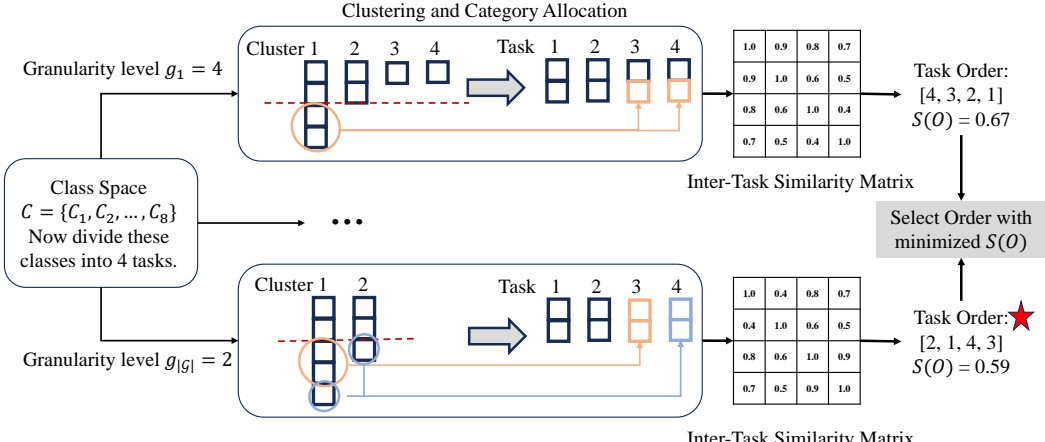

Figure A3: Details of generating difficult category sequences (Algorithm 1)

---

**Algorithm 1** Hard Task Sequence Generation Algorithm

---

**Require:** Similarity matrix $\mathbf{D}$, classes number $N$, tasks number $K$, candidate granularities set $\mathcal{G}$
**Ensure:** Task sequence $\mathcal{O}$
1: Initialize similarity graph $G \leftarrow \mathbf{D}$; set $G_{ii} \leftarrow 0$ for all $i$
2: Compute dissimilarity matrix $M \leftarrow \mathbf{I} - G$
3: **for** Granularity level $g \in \mathcal{G}$ **do**
4:      Perform hierarchical clustering on $M$ into $g$ clusters
5:      Merge clusters into $K$ tasks $\{\mathcal{C}_1, \ldots, \mathcal{C}_K\}$, minimizing cross-cluster task assignment
6:      Initialize inter-task similarity matrix $\mathbf{ITS} \in \mathbb{R}^{K \times K}$
7:      **for** $i, j \in [K], \ i \neq j$ **do**
8:          $\mathbf{ITS}_{ij} \leftarrow \frac{1}{|\mathcal{C}_i||\mathcal{C}_j|} \sum_{c_1 \in \mathcal{C}_i} \sum_{c_2 \in \mathcal{C}_j} G_{c_1 c_2}$
9:      **end for**
10:      Select the first task: $\mathcal{O}_g \leftarrow [\arg\min_{k \in [K]} \sum_{j \in [K], j \neq k} \mathbf{ITS}_{kj}]$
11:      Initialize remaining task set: $\mathcal{R} \leftarrow [K] \setminus \mathcal{O}_g$
12:      **while** $\mathcal{R} \neq \emptyset$ **do**
13:          Select the next task: $k^* \leftarrow \arg\min_{k \in \mathcal{R}} \sum_{t \in \mathcal{O}_g} \mathbf{ITS}_{tk}$
14:          Append $k^*$ to the task sequence: $\mathcal{O}_g \leftarrow \mathcal{O}_g \circ k^*$
15:          Update remaining task set: $\mathcal{R} \leftarrow \mathcal{R} \setminus \{k^*\}$
16:      **end while**
17:      Compute sequence score $\mathcal{S}(\mathcal{O}_g)$ according to Equation (30)
18: **end for**
19: Select the sequence: $\mathcal{O} \leftarrow \arg\min_{\mathcal{O}_g} \mathcal{S}(\mathcal{O}_g)$
20: **return** $\mathcal{O}$

---

- **Step 6-9 (Inter-Task Similarity Matrix):** The normalized average similarity $\mathbf{ITS}_{ij}$ accurately reflects task relationships as defined in Equation (30). This ensures algorithmic objectives align with theoretical similarity metrics.

- **Step 10-16 (Greedy Sequence Construction):** The initialization strategy selects the most isolated task as the starting point, preventing early error propagation. The iterative selection of least similar subsequent tasks implements a locally optimal strategy that approximates global minimization of $\mathcal{S}(\mathcal{O})$.

- **Step 17 (Multi-Granularity Optimization):** Evaluating multiple granularities leverages Equation (30), where better local minima are more likely to be found through diversified grouping strategies.

Figure A3 provides an illustrative example of the proposed procedure. Suppose we are given 8 classes to be partitioned into 4 tasks. Under a finer clustering granularity, the classes are grouped into 4

clusters, where the first cluster contains 4 classes, the second contains 3, and the remaining two contain 1 class each. To maintain high intra-task similarity, the two extra classes in the first cluster are redistributed to clusters 3 and 4, resulting in a balanced 4-task partition.

Under a coarser granularity, the same 8 classes might be clustered into only 2 groups: the first cluster with 5 classes and the second with 3. In this case, two of the most semantically similar classes from the larger cluster are assigned to form a new task, while the remaining two form another task, resulting in 4 tasks overall.

After generating task partitions, we compute the Inter-Task Similarity (ITS) matrix and select an initial task with the lowest global similarity. We then construct candidate sequences by greedily adding tasks with the smallest pairwise similarity to the most recently added task. For example, from this procedure, we may derive two sequences: $4 \to 3 \to 2 \to 1$ and $2 \to 1 \to 4 \to 3$, with corresponding similarity scores of 0.67 and 0.59, respectively. This process is repeated across clustering granularities, and the sequence with the lowest overall similarity score $\mathcal{S}(\mathcal{O})$ is ultimately selected as the hard sequence.

**Theorem 4** (Greedy Strategy Optimality Bound). *Let $\mathcal{O}_r$ be a uniformly random permutation of $K$ tasks and define the average inter-task similarity*

$$\bar{S} = \frac{1}{\binom{K}{2}} \sum_{1 \leq i < j \leq K} \frac{1}{|\mathcal{C}_i| \, |\mathcal{C}_j|} \sum_{c \in \mathcal{C}_i} \sum_{c' \in \mathcal{C}_j} Sim(c, c'), \tag{41}$$

*and assume $0 \leq Sim(c, c') \leq U$. Let $\mathcal{O}_g$ be the sequence produced by the greedy Algorithm 1. Then with probability at least $1 - e^{-K/2}$,*

$$\mathcal{S}(\mathcal{O}_g) \leq \mathbb{E}[\mathcal{S}(\mathcal{O}_r)] - \Delta, \tag{42}$$

*where*

$$\mathbb{E}[\mathcal{S}(\mathcal{O}_r)] = \frac{N^2 (K-1)}{2K^2} \bar{S}, \qquad \Delta = \frac{N^2 (K-1)}{2K^2} \bar{S} - \frac{2 N (\ln K + 1)}{K - 1} U. \tag{43}$$

*In particular, if $\bar{S} \geq \frac{4K^2 (\ln K + 1)}{N(K-1)^2} U$, then $\Delta > 0$ and hence $\mathcal{S}(\mathcal{O}_g) < \mathbb{E}[\mathcal{S}(\mathcal{O}_r)]$.*

*Proof.* Define, for $i < j$,

$$X_{i,j} = \sum_{c \in \mathcal{C}_i} \sum_{c' \in \mathcal{C}_j} Sim(c, c'). \tag{44}$$

Since each pair $(i, j)$ appears adjacent with probability $\frac{2}{K(K-1)}$,

$$\mathbb{E}[\mathcal{S}(\mathcal{O}_r)] = \sum_{i<j} \frac{2}{K(K-1)} X_{i,j} = \frac{2}{K(K-1)} \sum_{i<j} X_{i,j}. \tag{45}$$

Noting $|\mathcal{C}_i| = N/K$, one finds

$$\sum_{i<j} X_{i,j} = \binom{K}{2} \frac{N^2}{K^2} \bar{S} \implies \mathbb{E}[\mathcal{S}(\mathcal{O}_r)] = \frac{N^2(K-1)}{2K^2} \bar{S}. \tag{46}$$

At step $t$ of Algorithm 1, there are $t(K - t)$ candidate edges, each bounded by $N^2 U$. By standard order-statistic arguments,

$$\mathbb{E}[\Delta_t] \leq \frac{N^2 U}{t(K - t) + 1}, \tag{47}$$

and a union bound shows that with probability $\geq 1 - e^{-K/2}$ each $\Delta_t$ is at most twice its mean. Summing over $t = 1, \ldots, K - 1$ gives

$$\mathbb{E}[\mathcal{S}(\mathcal{O}_g)] \leq \frac{K}{(K-1)N} \sum_{t=1}^{K-1} \frac{N^2 U}{t(K-t)+1} \leq \frac{2 N (\ln K + 1)}{K - 1} U. \tag{48}$$

With probability at least $1 - e^{-K/2}$,

$$\mathcal{S}(\mathcal{O}_g) \leq 2 \mathbb{E}[\mathcal{S}(\mathcal{O}_g)] \leq \frac{4 N (\ln K + 1)}{K - 1} U. \tag{49}$$

Therefore

$$\mathbb{E}\big[\mathcal{S}(\mathcal{O}_r)\big] - \mathcal{S}(\mathcal{O}_g) \geq \frac{N^2(K-1)}{2K^2}\,\bar{S} - \frac{4\,N\,(\ln K + 1)}{K-1}\,U = \Delta, \qquad (50)$$

Completing the proof. □

Theorem 4 tells us that the greedy strategy of always choosing the most similar remaining pair of tasks takes advantage of strong inter-task affinities to produce an ordering whose total similarity remains tightly controlled and, when the average similarity is high enough, is lower than that of a random arrangement. The theorem shows that optimal local choices based only on current similarity scores accumulate into a reliable global solution even in noise.

**Complexity Analysis.** With a time complexity of $O(|\mathcal{G}|(N^3 + K^3))$, the algorithm remains tractable for practical CIL scenarios where $K \ll N$. The cubic terms stem primarily from hierarchical clustering (Step 4) and inter-task similarity computations (Step 6–9). In practice, these steps can be further accelerated using approximate nearest neighbor techniques. For instance, when partitioning 100 classes into 10 tasks, the algorithm completes in approximately **0.5 seconds**; for 200 classes into 10 tasks, it takes around **0.9 seconds** on a standard CPU, demonstrating its efficiency for common CIL settings.

Similarly, Algorithm 2 presents the pseudocode for constructing a simple task sequence by iteratively selecting and appending each task according to the prescribed rule.

---

**Algorithm 2** Easy Task Sequence Generation Algorithm

---

**Require:** Similarity matrix $\mathbf{D}$, classes number $N$, tasks number $K$, candidate granularities set $\mathcal{G}$
**Ensure:** Task sequence $\mathcal{O}$
 1: Initialize similarity graph $G \leftarrow I - \mathbf{D}$; set $G_{ii} \leftarrow 0$ for all $i$
 2: Compute dissimilarity matrix $M \leftarrow \mathbf{I} - G$
 3: **for** Granularity level $g \in \mathcal{G}$ **do**
 4:     Perform hierarchical clustering on $M$ into $g$ clusters
 5:     Merge clusters into $K$ tasks $\{\mathcal{C}_1, \ldots, \mathcal{C}_K\}$, minimizing cross-cluster task assignment
 6:     Initialize inter-task similarity matrix $\mathbf{ITS} \in \mathbb{R}^{K \times K}$
 7:     **for** $i, j \in [K]$, $i \neq j$ **do**
 8:         $\mathbf{ITS}_{ij} \leftarrow \frac{1}{|\mathcal{C}_i||\mathcal{C}_j|} \sum_{c_1 \in \mathcal{C}_i} \sum_{c_2 \in \mathcal{C}_j} G_{c_1 c_2}$
 9:     **end for**
10:     Select the first task: $\mathcal{O}_g \leftarrow [\arg\max_{k \in [K]} \sum_{j \in [K], j \neq k} \mathbf{ITS}_{kj}]$
11:     Initialize remaining task set: $\mathcal{R} \leftarrow [K] \setminus \mathcal{O}_g$
12:     **while** $\mathcal{R} \neq \emptyset$ **do**
13:         Select the next task: $k^* \leftarrow \arg\max_{k \in \mathcal{R}} \sum_{t \in \mathcal{O}_g} \mathbf{ITS}_{tk}$
14:         Append $k^*$ to the task sequence: $\mathcal{O}_g \leftarrow \mathcal{O}_g \circ k^*$
15:         Update remaining task set: $\mathcal{R} \leftarrow \mathcal{R} \setminus \{k^*\}$
16:     **end while**
17:     Compute sequence score $\mathcal{S}(\mathcal{O}_g)$ according to Equation (30)
18: **end for**
19: Select the sequence: $\mathcal{O} \leftarrow \arg\max_{\mathcal{O}_g} \mathcal{S}(\mathcal{O}_g)$
20: **return** $\mathcal{O}$

---

# D    DETAILED ANALYSIS OF EXPERIMENT

## D.1    ENUMERABLE EXPERIMENTS

### D.1.1    EXPERIMENTAL SETUP

**Dataset and Metrics.** We conduct experiments on two standard benchmarks: CIFAR-100 Krizhevsky (2009) and ImageNet-R Krizhevsky (2009). For each, we select the first six semantic classes and group them into three sequential learning tasks of two classes each, yielding a total of $3! = 6$ possible task orders per dataset; by considering all class-to-task assignments, we obtain 90 distinct sequences, which we treat as the ground-truth distribution $\mathcal{D}_{\text{true}}$. To estimate this distribution in practice, we use:

- **Random Seed (RS) protocol:** draw task sequences by shuffling class-labels under random seeds $\{0, 42, 1993\}$ Lai et al. (2025); Li & Zhou (2025); McDonnell et al. (2024); Wang et al. (2022e), forming the empirical distribution $\mathcal{D}_{\mathrm{RS}}$.
- **EDGE protocol:** apply our edge-selection strategy on the same seeds to produce $\mathcal{D}_{\mathrm{EDGE}}$.

We compare each estimated distribution to $\mathcal{D}_{\mathrm{true}}$ using two complementary divergence metrics:

- **Jensen–Shannon divergence** $JSD_d$: Given two discrete distributions $P$ and $Q$ over the same support, the Jensen–Shannon divergence is defined as

$$JSD(P \parallel Q) = \tfrac{1}{2} D_{\mathrm{KL}}\big(P \parallel M\big) + \tfrac{1}{2} D_{\mathrm{KL}}\big(Q \parallel M\big), \quad M = \tfrac{1}{2}(P + Q),$$

  where $D_{\mathrm{KL}}$ is the Kullback–Leibler divergence. Unlike $D_{\mathrm{KL}}$, the $JSD$ is symmetric and bounded in $[0, \ln 2]$, which makes it well suited for measuring similarity between empirical distributions with potentially non-overlapping support Lamberti et al. (2007).
- **Wasserstein distance** $W_d$: Also known as the Earth Mover's Distance, the first-order Wasserstein distance between $P$ and $Q$ on a metric space $(\mathcal{X}, d)$ is

$$W_1(P, Q) = \inf_{\gamma \in \Gamma(P,Q)} \mathbb{E}_{(x,y) \sim \gamma}\big[d(x, y)\big],$$

  where $\Gamma(P, Q)$ denotes the set of all joint distributions with marginals $P$ and $Q$. In the discrete case, this reduces to the minimum cost of transporting "mass" from $P$ to $Q$, providing a meaningful measure of distributional distance that accounts for the geometry of the task-permutation space Villani (2009).

**Implementation Details.** All methods are implemented in PyTorch with the following shared hyperparameters:

- **Memory:** total size 2000, up to 20 samples per class, non-fixed allocation.
- **Backbone:** ResNet-18, trained from scratch in the non-pre-trained setting and with ImageNet pre-training otherwise.
- **Optimizer & Scheduler:** SGD with step-LR; initial learning rate 0.1, weight decay $5 \times 10^{-4}$, LR decay factor 0.1 at epochs $\{60, 120, 170\}$ (non-pre-trained) or $\{80, 120, 150\}$ (pre-trained).
- **Training:** 170 epochs, batch size 128.

### D.1.2   ADDITIONAL EXPERIMENT RESULTS

Table A4: Fitting performance of EDGE on CIFAR-100 using HidePrompt with various backbones. **EDGE consistently outperforms RS across nearly all backbones, demonstrating its effectiveness and robustness to different model architectures.**

| Model | $JSD_d$ | $W_d$ |
|---|---|---|
| RS Estimate | 0.2694 | 2.8688 |
| EDGE with ResNet50 | 0.0863 | 1.5677 |
| EDGE with ResNet50×64 | 0.1986 | 3.2553 |
| EDGE with ResNet101 | 0.1386 | 1.7020 |
| EDGE with ViT-B/16 | 0.1236 | 1.5196 |
| EDGE with ViT-B/32 | 0.1237 | 2.3599 |
| EDGE with ViT-L/14 | **0.0846** | **1.0642** |

Table A4 summarizes the fitting performance of the EDGE method on the CIFAR-100 dataset under the HidePrompt setting, using various backbone architectures. The performance is evaluated in terms of Jensen-Shannon Divergence ($JSD_d$) and the 2-Wasserstein distance ($W_d$). Among all configurations, EDGE with ViT-L/14 best fits the reference distribution, yielding the lowest $JSD_d$ (0.0846 bits) and the smallest $W_d$ distance (1.0642).

Figure A4 and Figure A5 visualize the ground-truth performance distributions (black), along with the estimates produced by the RS protocol (blue) and our proposed EDGE protocol (red), for non-pre-trained and pre-trained CIL methods, respectively. These results demonstrate EDGE's superior ability to approximate the true distribution, capturing both the central tendency and the spread more accurately than the conventional RS protocol.

### D.1.3 DISCUSSION ON EDGE FOR MODEL SELECTION

In addition to providing a more reliable evaluation, EDGE offers new insights for model selection. To demonstrate this, we compare continual learning method rankings under EDGE and RS across three dimensions: performance upper bound, performance lower bound, and stability, and quantify the consistency between the two evaluation protocols using ranking distance.

Table A5 presents the rankings under the fixed-class setting described in Section 5.1. We observe that EDGE rankings are overall closer to the reference ordering across all three dimensions, while RS exhibits larger deviations, resulting in higher total ranking errors. Specifically, on CIFAR-100, EDGE's ranking error is 6 compared to 12 for RS, and on ImageNet-R, 2 versus 10.

Table A5: Model rankings derived from Table 1. The reference (true) ranking is highlighted in gray.

| Method \ Rank | CIFAR-100 | | | | | | | | | ImageNet-R | | | | | | | | |
|---|---|---|---|---|---|---|---|---|---|---|---|---|---|---|---|---|---|---|
| | Lower Bound | | | Upper Bound | | | Stability | | | Lower Bound | | | Upper Bound | | | Stability | | |
| | Real | EDGE | RS | Real | EDGE | RS | Real | EDGE | RS | Real | EDGE | RS | Real | EDGE | RS | Real | EDGE | RS |
| L2P | 5 | 6 | 4 | 4 | 4 | 5 | 5 | 5 | 5 | 5 | 5 | 5 | 4 | 4 | 6 | 5 | 5 | 6 |
| CODA-Prompt | 6 | 5 | 6 | 6 | 6 | 6 | 6 | 6 | 5 | 6 | 6 | 6 | 6 | 6 | 6 | 6 | 6 | 5 |
| Hide-Prompt | 4 | 4 | 5 | 5 | 5 | 4 | 4 | 3 | 6 | 4 | 4 | 4 | 5 | 5 | 4 | 4 | 3 | 1 |
| EASE | 2 | 2 | 2 | 3 | 3 | 3 | 2 | 1 | 1 | 2 | 2 | 2 | 3 | 3 | 2 | 1 | 1 | 2 |
| MOS | 3 | 3 | 3 | 2 | 2 | 2 | 3 | 4 | 3 | 3 | 3 | 3 | 2 | 2 | 3 | 3 | 4 | 3 |
| RanPAC | 1 | 1 | 1 | 1 | 1 | 1 | 1 | 2 | 2 | 1 | 1 | 1 | 1 | 1 | 1 | 2 | 2 | 4 |

To further test the robustness of these findings, we repeated the experiment using a randomly sampled set of six classes (seed 42), as reported in Table A6. EDGE again outperforms RS: on CIFAR-100, EDGE achieves a ranking error of 0 versus 12 for RS; on ImageNet-R, 2 versus 12. These results confirm that EDGE consistently produces rankings that are closer to the reference ordering, providing a more reliable basis for model selection.

Table A6: Model rankings obtained using a randomly selected set of classes (seed 42). The reference (true) ranking is highlighted in gray.

| Method \ Rank | CIFAR-100 | | | | | | | | | ImageNet-R | | | | | | | | |
|---|---|---|---|---|---|---|---|---|---|---|---|---|---|---|---|---|---|---|
| | Lower Bound | | | Upper Bound | | | Stability | | | Lower Bound | | | Upper Bound | | | Stability | | |
| | Real | EDGE | RS | Real | EDGE | RS | Real | EDGE | RS | Real | EDGE | RS | Real | EDGE | RS | Real | EDGE | RS |
| L2P | 5 | 5 | 3 | 1 | 1 | 1 | 6 | 6 | 5 | 5 | 5 | 5 | 3 | 3 | 5 | 5 | 5 | 4 |
| CODA-Prompt | 6 | 6 | 5 | 5 | 5 | 5 | 4 | 4 | 4 | 6 | 6 | 6 | 6 | 6 | 6 | 6 | 6 | 6 |
| Hide-Prompt | 4 | 4 | 6 | 4 | 4 | 4 | 5 | 5 | 6 | 4 | 4 | 5 | 5 | 5 | 4 | 4 | 4 | 2 |
| EASE | 3 | 3 | 4 | 6 | 6 | 6 | 3 | 3 | 1 | 2 | 2 | 2 | 1 | 2 | 2 | 2 | 2 | 1 |
| MOS | 2 | 2 | 2 | 3 | 3 | 3 | 1 | 1 | 2 | 3 | 3 | 3 | 4 | 4 | 3 | 3 | 3 | 3 |
| RanPAC | 1 | 1 | 1 | 2 | 2 | 2 | 2 | 2 | 3 | 1 | 1 | 1 | 2 | 1 | 1 | 1 | 1 | 5 |

**Conclusion.** Across both fixed-class and random-class settings, EDGE demonstrates superior fidelity in reflecting the true performance ordering of continual learning methods. It more accurately captures worst-case robustness, best-case potential, and stability—properties critical for dependable deployment in practical scenarios. Overall, these findings highlight the value of distribution-aware evaluation and demonstrate that EDGE provides more informative guidance for continual learning model selection than RS.

### D.2 ANALYSIS OF LARGE-SCALE EXPERIMENT

Table A7 and Table A8 present the evaluation results of existing CIL methods under both RS and EDGE protocols. Notably, we observe conclusions consistent with those discussed in Section 5.1. From the perspective of EDGE, these results offer new insights into CIL model design and selection:

- **The realistic performance range of CIL models can be substantially wider than what is captured by RS protocols.** EDGE effectively identifies both easy and challenging class sequences in most cases, and demonstrates broad applicability across pre-trained and non-pre-trained models. For example, on the CUB dataset, the performance range of L2P expands from 2.38 to 16.25, while that of TagFex increases from 1.06 to 7.27, enabling a more accurate and nuanced understanding of model behavior. These findings highlight the importance of considering extreme task sequences during model design to ensure robustness under diverse deployment scenarios.

Table A7: Performance of pre-trained model-based CIL methods under two evaluation protocols. **White background denotes the RS protocol, while gray background denotes the EDGE protocol.** Reported are the sampled minimum and maximum accuracies, along with the estimated mean and standard deviation of the ground truth performance distribution (unit: %).

| Method | CIFAR100 | | | CUB | | | ImageNet-R | | |
|---|---|---|---|---|---|---|---|---|---|
| | $\min_{\mathcal{A}_N}$ | $\max_{\mathcal{A}_N}$ | $\mu_{\mathcal{A}_N}\pm\sigma_{\mathcal{A}_N}$ | $\min_{\mathcal{A}_N}$ | $\max_{\mathcal{A}_N}$ | $\mu_{\mathcal{A}_N}\pm\sigma_{\mathcal{A}_N}$ | $\min_{\mathcal{A}_N}$ | $\max_{\mathcal{A}_N}$ | $\mu_{\mathcal{A}_N}\pm\sigma_{\mathcal{A}_N}$ |
| L2P | 82.93 | 84.48 | $83.46_{\pm0.72}$ | 66.18 | 68.56 | $67.25_{\pm0.99}$ | 71.10 | 71.43 | $71.29_{\pm0.14}$ |
| | 81.67 | 84.62 | $83.08_{\pm1.21}$ | 59.75 | 76.00 | $67.31_{\pm6.68}$ | 71.02 | 72.37 | $71.61_{\pm0.57}$ |
| CODA-Prompt | 85.17 | 85.56 | $85.30_{\pm0.18}$ | 70.32 | 71.56 | $70.74_{\pm0.59}$ | 73.03 | 73.54 | $73.27_{\pm0.21}$ |
| | 84.82 | 85.65 | $85.22_{\pm0.34}$ | 67.83 | 76.42 | $71.94_{\pm3.52}$ | 70.80 | 74.50 | $72.95_{\pm1.57}$ |
| Hide-Prompt | 85.06 | 86.20 | $85.45_{\pm0.53}$ | 81.69 | 82.77 | $82.06_{\pm0.50}$ | 71.76 | 73.16 | $72.58_{\pm0.60}$ |
| | 84.25 | 87.36 | $85.56_{\pm1.32}$ | 80.49 | 85.44 | $82.90_{\pm2.02}$ | 70.75 | 72.83 | $71.79_{\pm0.85}$ |
| RanPAC | 90.32 | 90.87 | $90.68_{\pm0.25}$ | 89.34 | 89.75 | $89.49_{\pm0.19}$ | 77.27 | 77.32 | $77.30_{\pm0.02}$ |
| | 90.25 | 90.87 | $90.65_{\pm0.29}$ | 89.31 | 89.90 | $89.66_{\pm0.25}$ | 75.97 | 77.65 | $76.97_{\pm0.72}$ |
| EASE | 87.24 | 87.53 | $87.35_{\pm0.13}$ | 81.09 | 83.06 | $82.21_{\pm0.82}$ | 75.89 | 76.12 | $76.00_{\pm0.09}$ |
| | 85.77 | 88.41 | $87.15_{\pm1.08}$ | 81.56 | 83.33 | $82.45_{\pm0.72}$ | 75.46 | 75.97 | $75.79_{\pm0.23}$ |
| MOS | 90.69 | 91.22 | $91.03_{\pm0.24}$ | 88.87 | 89.39 | $89.08_{\pm0.23}$ | 76.90 | 77.33 | $77.15_{\pm0.18}$ |
| | 90.79 | 91.22 | $91.01_{\pm0.18}$ | 87.69 | 90.16 | $89.08_{\pm1.03}$ | 76.48 | 77.93 | $77.21_{\pm0.59}$ |

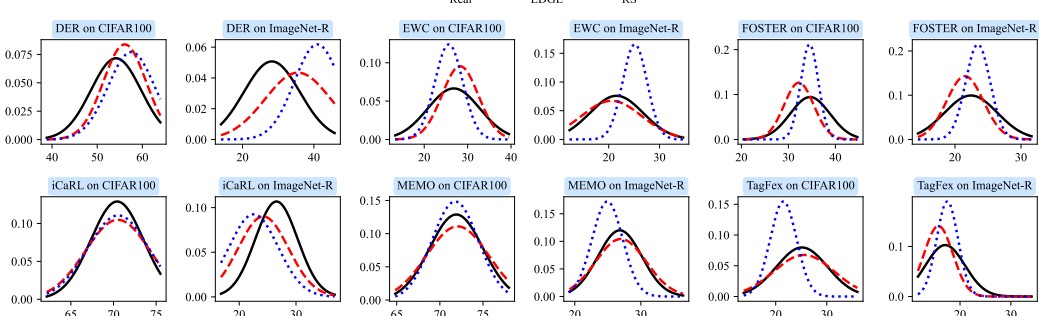

Figure A4: The ground-truth distribution (black), along with estimates from the EDGE protocol (red) and the RS protocol (blue), for non–pre-trained CIL methods. EDGE provides a more faithful approximation to the true performance distribution.

- **Model rankings may change under extreme task sequences.** For example, on the CUB dataset, MOS and RanPAC exhibit comparable performance under the RS protocol, yet diverge significantly when evaluated with EDGE: MOS attains a higher upper bound (up to 90.16) but experiences a notable drop in its lower bound. This indicates that algorithm selection should be informed by specific deployment priorities, whether emphasizing worst-case robustness or maximizing best-case accuracy. EDGE offers valuable empirical evidence to support such scenario-aware decision-making.

- **Some model designs exhibit inherent limitations.** For instance, on the ImageNet-R dataset, the lower bounds of three prompt-based methods all approach 70%, indicating that certain difficult sequences can drastically undermine their effectiveness. This observation suggests that analyzing which types of sequences consistently degrade performance can help identify structural weaknesses in different methods. Such insights can inform targeted improvements in model robustness, guide the development of sequence-aware training strategies, and support the selection of appropriate models for deployment in challenging real-world scenarios.

### D.3  ANALYSIS OF CLIP ENCODING

When designing the EDGE protocol, our objective is to construct representative class sequences of varying difficulty without accessing actual image instances. To this end, we employ the CLIP text encoder, a vision-language model trained with contrastive learning that exhibits strong visual-text alignment and zero-shot generalization capabilities.

Table A8: Performance of non-pre-trained CIL methods under two evaluation protocols. Other notations follow those in Table A7.

| Method | CIFAR100 | | | CUB | | | ImageNet-R | | |
|---|---|---|---|---|---|---|---|---|---|
| | $\min_{\mathcal{A}_N}$ | $\max_{\mathcal{A}_N}$ | $\mu_{\mathcal{A}_N} \pm \sigma_{\mathcal{A}_N}$ | $\min_{\mathcal{A}_N}$ | $\max_{\mathcal{A}_N}$ | $\mu_{\mathcal{A}_N} \pm \sigma_{\mathcal{A}_N}$ | $\min_{\mathcal{A}_N}$ | $\max_{\mathcal{A}_N}$ | $\mu_{\mathcal{A}_N} \pm \sigma_{\mathcal{A}_N}$ |
| EWC | 13.83 | 14.94 | $14.34_{\pm0.45}$ | 10.31 | 10.90 | $10.60_{\pm0.24}$ | 7.38 | 7.77 | $7.61_{\pm0.17}$ |
| | 13.79 | 17.22 | $15.08_{\pm1.52}$ | 8.18 | 10.31 | $9.46_{\pm0.92}$ | 5.47 | 7.79 | $7.01_{\pm1.09}$ |
| DER | 57.59 | 59.73 | $58.48_{\pm0.91}$ | 46.65 | 48.05 | $47.50_{\pm0.61}$ | 29.37 | 32.92 | $31.57_{\pm1.57}$ |
| | 56.42 | 60.20 | $58.25_{\pm1.54}$ | 45.25 | 49.62 | $47.17_{\pm1.82}$ | 29.28 | 34.92 | $32.37_{\pm2.33}$ |
| iCaRL | 36.60 | 41.54 | $38.85_{\pm2.03}$ | 32.10 | 32.57 | $32.36_{\pm0.19}$ | 15.43 | 16.40 | $15.83_{\pm0.41}$ |
| | 34.16 | 40.56 | $37.11_{\pm2.63}$ | 30.40 | 35.28 | $32.59_{\pm2.02}$ | 13.55 | 16.78 | $15.58_{\pm1.44}$ |
| FOSTER | 48.43 | 51.47 | $49.87_{\pm1.25}$ | 42.66 | 43.75 | $43.07_{\pm0.49}$ | 18.70 | 20.52 | $19.47_{\pm0.77}$ |
| | 49.21 | 51.23 | $49.62_{\pm1.17}$ | 38.25 | 45.25 | $42.42_{\pm3.01}$ | 17.03 | 21.52 | $19.25_{\pm1.83}$ |
| MEMO | 55.16 | 58.49 | $56.80_{\pm1.36}$ | 39.31 | 41.52 | $40.23_{\pm0.94}$ | 20.05 | 21.70 | $21.08_{\pm0.73}$ |
| | 54.96 | 58.96 | $56.36_{\pm1.84}$ | 39.31 | 41.31 | $40.19_{\pm0.83}$ | 19.50 | 21.70 | $20.87_{\pm0.98}$ |
| TagFex | 62.23 | 62.69 | $62.42_{\pm0.19}$ | 46.06 | 47.12 | $46.47_{\pm0.46}$ | 34.05 | 34.27 | $34.16_{\pm0.09}$ |
| | 60.78 | 68.80 | $63.94_{\pm3.48}$ | 42.62 | 49.89 | $46.25_{\pm2.97}$ | 33.38 | 34.72 | $34.05_{\pm0.55}$ |

Figure A5: The ground-truth distribution (black), and the corresponding estimates from EDGE (red) and RS (blue) protocols, for pre-trained CIL methods. The results highlight the improved accuracy of EDGE in capturing both the central tendency and variance.

### D.3.1 DISCUSSION ON NATURAL IMAGES.

A natural concern is whether the semantic similarity measured by CLIP text embeddings faithfully reflects the actual visual difficulty among classes. To examine this, we compare the similarity matrix obtained from the CLIP text encoder with the similarity matrix derived from real image features. We evaluate the results using two metrics:

- **Mean Absolute Error (MAE)**: quantifies the average deviation between the similarity matrices computed from CLIP embeddings and image features. Lower values indicate closer alignment.
- **Consistency of Sequence Generation (CSG)**: measures the proportion of classes that remain assigned to the same task after generating sequences using the two similarity matrices. Higher values indicate more stable sequence generation.

Table A9: Comparison between CLIP text-based similarity and image-based similarity across datasets.

| Dataset | CIFAR100 | CUB | ImageNet-R |
|---|---|---|---|
| MAE ($\downarrow$) | 0.14 | 0.08 | 0.17 |
| CSG ($\uparrow$) | 0.79 | 0.83 | 0.77 |

The results indicate that although a moderate deviation exists between semantic and image similarities (MAE around $0.15$), CLIP embeddings capture the underlying relational structure effectively. In particular, CSG remains consistently above $0.75$, showing that the generated extreme sequences are robust and align well with those derived from image-based similarities. This confirms the feasibility of using CLIP to measure similarity for generating extreme sequences within EDGE.

### D.3.2 DISCUSSION ON NON-NATURAL/PROFESSIONAL IMAGES

A potential issue is whether CLIP text embeddings derived from class names faithfully capture visual relationships in specialized, non-natural domains. This is a nontrivial problem for several reasons:

- Many domain-specific class labels are terse, technical, or stage-based (e.g., "moderate" vs "severe") and therefore omit visual descriptors such as color, texture, or morphology that are essential for visual discrimination.
- Within-class heterogeneity and between-class subtlety are common: distinct clinical labels can correspond to overlapping or gradual visual features (e.g., small hemorrhages vs. microaneurysms), making semantic labels a poor proxy for perceptual distance.
- Dataset issues such as class imbalance, labeling protocol differences and inter-observer variability further weaken the simple mapping from a short class name to an image-space distribution.

Together, these factors explain why directly using bare class names with a general-purpose text encoder can fail to reflect true image-space similarity in professional domains.

To assess this empirically, we examined two representative medical-image benchmarks. **EyePACS** is a large-scale retinal fundus dataset for diabetic retinopathy grading, containing color fundus photographs acquired with diverse cameras and imaging conditions. Visual cues range from microaneurysms and small hemorrhages to hard exudates and neovascularization. Labels correspond to five DR severity levels (no, mild, moderate, severe, proliferative), which encode stage progression rather than detailed appearance descriptors. **HAM10000** is a dermatoscopic image dataset of pigmented skin lesions collected from multiple clinical sources. Images exhibit substantial variability in morphology, color, and acquisition artifacts, and several diagnostic categories are visually similar. The seven classes used here are Actinic keratosis, Basal cell carcinoma, Benign keratosis, Dermatofibroma, Melanoma, Nevi, and Vascular lesion.

For each dataset we constructed (a) an inter-class similarity matrix from CLIP text embeddings of class names and (b) an inter-class similarity matrix from image prototypes. We measured agreement between (a) and (b) using Spearman's rank correlation. Using class names directly yields only modest alignment with image-derived similarities: EyePACS shows Spearman's $\rho = 0.588$ (p $\approx 0.07$), and HAM10000 shows $\rho = 0.279$ (p $\approx 0.22$), consistent with the intuition above that short, technical labels do not reliably encode visual detail in these domains.

To increase the visual content of the textual representations, we expanded each class name into a concise, visually informative caption using a large language model (GPT-5 in this experiment). The prompt we used is shown below inside a boxed, two-end–justified block:

> **Prompt Template**
>
> ```
> Generate one concise, visually descriptive caption (8-20
> words) that highlights the typical visual appearance, color,
> texture, and anatomical context of a {class_name} lesion in
> medical images.
> ```

We encoded the generated captions with CLIP and recomputed the class similarity matrices. This simple augmentation substantially increased agreement: EyePACS improved to Spearman's $\rho = 0.863$ (p $\approx 0.01$), and HAM10000 improved to $\rho = 0.653$ (p $\approx 0.02$). These results indicate that short, visually focused textual expansions recover much of the image-space relational structure that bare class names miss.

### D.4 DISCUSSION OF OTHER POTENTIAL BASELINES

Although most CIL evaluations use the RS protocol, comparing only to RS risks underestimating EDGE's ability to find challenging task sequences. We therefore include several additional, conceptually distinct baselines to evaluate both effectiveness and efficiency:

- **LLM-generated sequences. LLM-1:** A single-round generation procedure in which a large language model directly produces a candidate sequence based on a prompt describing "easy" or

"hard" sequences. This baseline tests whether semantic difficulty can be inferred directly from class names without any iterative refinement; $\approx$130 s per sequence. **LLM-5:** A five-round iterative refinement procedure. Each round, the LLM receives feedback regarding the previously generated sequence and attempts to correct or adjust its output in the next iteration. This baseline evaluates whether multi-step reasoning helps the LLM better capture difficulty; $\approx$600 s per sequence.

- **Adversarial Sampling (AS).** A greedy, similarity-based adversarial strategy. At each step, AS selects the class that is maximally dissimilar from all currently selected classes, thereby increasing sequence difficulty by pushing the sequence toward the tail of the similarity distribution; $\approx$0.9 s per sequence.
- **Max-cover Sampling (MS).** A randomized search-based approach. We first sample a pool of candidate sequences (we use 200), compute for each sequence a coverage or farthest-distance score relative to previously selected sets, and finally choose the top-ranked sequences; $\approx$8 s per sequence.

Table A10: Comparison of EDGE against alternative baselines (Hard / Easy correspond to sequences intended to be difficult / easy for the evaluated methods).

| Method | EDGE | | RS | | AS | | MS | | LLM-1 | | LLM-5 | |
|---|---|---|---|---|---|---|---|---|---|---|---|---|
| | Hard | Easy | Hard | Easy | Hard | Easy | Hard | Easy | Hard | Easy | Hard | Easy |
| L2p | **58.35** | **72.13** | 62.58 | 66.21 | 61.75 | 69.48 | 62.60 | 63.70 | 64.97 | 66.92 | 67.01 | 67.68 |
| CODA-Prompt | **65.65** | 69.42 | 67.42 | 68.12 | 65.96 | 67.33 | 66.13 | 68.18 | 67.01 | 68.58 | 65.78 | **69.59** |
| Hide-Prompt | **80.52** | **83.21** | 81.45 | 82.35 | 81.02 | 82.39 | 80.96 | 82.63 | 82.35 | 82.36 | 80.89 | 81.45 |
| RanPAC | **88.68** | **89.40** | 88.72 | 89.15 | 88.99 | 89.21 | 88.72 | 89.21 | 88.72 | 89.25 | 88.72 | 88.68 |
| EASE | **84.29** | **85.37** | 84.60 | 84.96 | 84.69 | 85.07 | 84.39 | 85.33 | 84.56 | 85.01 | 84.82 | 84.78 |
| MOS | **87.69** | **89.56** | 88.49 | 88.98 | 88.76 | 88.93 | 88.38 | 88.30 | 89.13 | 88.56 | 88.23 | 89.26 |

**Key observations.** Two main conclusions follow from these comparisons:

1. **LLM-based generation is effective but costly and unstable.** Single-shot LLM outputs (LLM-1) are highly variable and frequently fail to produce consistently hard sequences. Iterative prompting (LLM-5) significantly improves stability and often surpasses RS, but remains substantially less effective than EDGE in most cases. Crucially, multi-turn LLM workflows are orders of magnitude slower (a single interactive round typically takes 2–3 minutes in our setup; five rounds commonly exceed 10 minutes), making them impractical for large-scale or low-latency evaluation.
2. **Sampling-based methods find some hard cases but lack transferability.** AS and MS are computationally efficient and can locate sequences that are challenging for particular algorithms. However, the difficult sequences they discover frequently exploit idiosyncrasies of a single target method and do not generalize across the range of CIL approaches we evaluate. By contrast, EDGE identifies extreme sequences that consistently increase difficulty across many methods, achieving a better balance of effectiveness and generality.

Overall, these results show that while alternative strategies can occasionally produce challenging sequences, EDGE provides the most reliable combination of performance, generality, and computational efficiency for discovering extreme task orders.

## D.5 DISCUSSION ON THE NUMBER OF SAMPLES

In previous sections we compared EDGE with the standard CIL evaluation protocol that uses RS by drawing three random task orders. In this subsection we simulate scenarios with an increased number of RS samples in order to investigate how RS-based evaluation behaves as the sample budget grows, and to further demonstrate the practical advantages of EDGE.

**Distribution estimation in the enumerable setting** The setup is described in Section 5.1. Figure A6 visualizes how the evaluation outcomes change for RS and EDGE as the number of sampled sequences increases. From these experiments we draw two main observations:

1. If only RS sampling is increased, RS reaches the distribution estimation quality produced by EDGE with three EDGE samples after roughly five to six RS samples. Note that the total sequence space in this experiment is only 90 sequences. In this limited space RS therefore requires about twice as many random sequences to match the estimation quality of EDGE.

2. If we increase the number of samples for both RS and EDGE, EDGE remains superior throughout. As the sample counts grow, the two procedures tend to converge, and this convergence typically occurs when the number of samples is on the order of ten to twenty sequences.

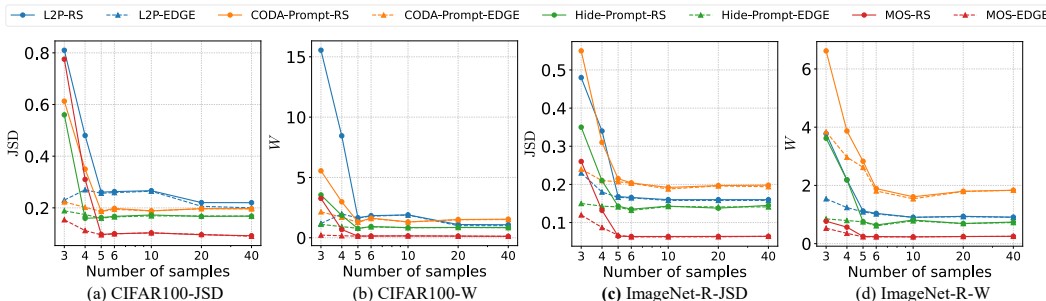

Figure A6: Evolution of RS and EDGE performance as the number of sampled sequences increases on CIFAR-100 and ImageNet-R. Circles denote RS data points and triangles denote EDGE data points. Different colored curves correspond to different pretrained continual learning methods.

**Extreme-sequence capture in classic CIL settings**   The setup for these experiments is described in Section 5.2. In classic class incremental learning settings the class space is typically very large, which prevents us from directly estimating the full performance distribution. Accordingly, we evaluate an evaluation protocol by its ability to capture extreme sequences. We performed a focused empirical study on the CUB dataset using two representative methods: L2P, which exhibits a wide performance range over sequences, and EASE, which exhibits a relatively narrow performance range.

Figure A7 presents the empirical distributions obtained by repeated RS sampling together with the single-shot EDGE positions. The specific observations are as follows:

- For L2P, under our setup EDGE found a hard-case accuracy of 58.5 and an easy-case accuracy of 72.3. We ran 600 RS samplings. Only 6 of those RS samples produced lower accuracies, with the minimum RS accuracy equal to 57.55. None of the RS samples attained an accuracy higher than EDGE.
- For EASE, under our setup EDGE found a hard-case accur acy of 84.29 and an easy-case accuracy of 85.37. After more than 400 random evaluations, only 4 RS samples produced lower accuracies, with minimum RS accuracy equal to 84.09. Only one RS sample achieved a higher accuracy, equal to 85.41.

These results confirm the intuitive fact that increasing RS sample count improves RS. However, such improvement typically entails a large evaluation cost. Compared with EDGE, discovering extreme sequences using only RS is both time consuming and inefficient. The empirical evidence therefore supports the practical value of EDGE as a more sample efficient and reliable procedure for identifying extreme task orders.

**Remarks**   All experimental details and plotting scripts are provided in Appendix E.5. The plots in Figure A6 and Figure A7 support the conclusion that EDGE achieves comparable or better distribution estimation with far fewer samples than RS, which reduces evaluation cost and improves the reliability of worst-case and best-case assessments.

## E   HOW TO USE THE EDGE REPOSITORY

We release an official implementation of EDGE for reproducing the proposed evaluation protocol in CIL.[1] Unlike earlier versions that only provided pre-generated class orders, the current repository integrates EDGE directly into two widely used CIL toolboxes, enabling users to switch between the standard random-order evaluation and the EDGE protocol via a unified command-line interface.

---

[1] https://github.com/AIGNLAI/EDGE

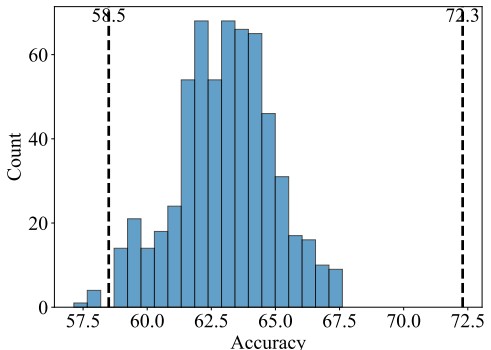 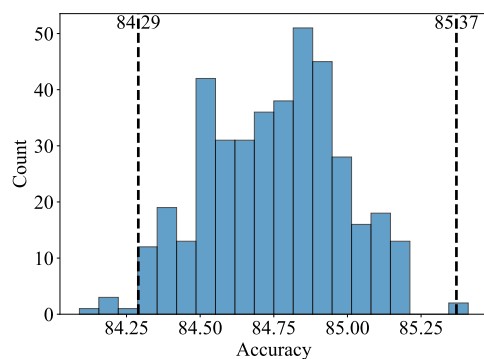

(a) Distribution of accuracies for 600 random sequences under L2P. The vertical black line marks the accuracy position obtained by a single EDGE sample.

(b) Distribution of accuracies for 400 random sequences under EASE. The vertical black line marks the accuracy position obtained by a single EDGE sample.

Figure A7: Random sampling distributions and EDGE single-shot positions on the CUB dataset. Left panel corresponds to L2P and right panel corresponds to EASE.

### E.1 REPOSITORY LAYOUT

The repository contains two integrated codebases:

- `PILOT/`: an EDGE-integrated fork based on the PILOT toolbox.
- `PyCIL/`: an EDGE-integrated fork based on the PyCIL toolbox.

In both integrations, EDGE is implemented as a modular evaluation component and can be invoked without changing the training recipe or model definition. This design keeps EDGE largely decoupled from the training pipeline, making it straightforward to apply to different methods supported by the toolboxes.

### E.2 INSTALLATION

We recommend creating a dedicated environment and installing the dependencies required by the corresponding toolbox. In addition to common CIL dependencies (PyTorch, torchvision, `numpy`, `tqdm`, etc.), EDGE requires the CLIP package for computing the class-name similarity used by the protocol.

```
git clone https://github.com/AIGNLAI/EDGE
cd EDGE

conda create -n edge python=3.10 -y
conda activate edge

pip install torch torchvision
pip install git+https://github.com/openai/CLIP.git
pip install scipy
pip install timm==0.6.12
pip install tqdm
```

### E.3 RUNNING: RANDOM PROTOCOL VS. EDGE PROTOCOL

EDGE is exposed via a single command-line flag (denoted as `-eval` in the repository usage guide) to select the evaluation protocol:

- `-eval random`: the standard random class-order protocol (randomly sampled class orders/seeds).

- `-eval edge`: the proposed EDGE protocol (adaptive sampling with extreme sequences to better approximate the performance distribution boundary).

A typical invocation is:

```
# Run with EDGE protocol
python main.py --config ./exps/[MODEL_NAME].json --eval edge

# Run with standard random-order protocol
python main.py --config ./exps/[MODEL_NAME].json --eval random
```

**Notes.** (1) **Config files.** The `-config` argument points to the experiment configuration provided by the toolbox (e.g., model, dataset, task size, memory budget, optimizer, and schedule). EDGE does not require modifying these training settings. (2) **Fair comparison.** For a fair comparison between protocols, keep the training configuration fixed and only switch `-eval`. If you use multiple seeds/orders under the random protocol, we recommend using the same overall evaluation budget when running EDGE. (3) **Outputs.** The scripts report the standard CIL metrics supported by the toolbox (e.g., average incremental accuracy). Under EDGE, the evaluation additionally emphasizes boundary-aware statistics by considering extreme sequences, which helps reveal the variance and worst/best-case behaviors that can be under-estimated by random sampling.

### E.4 PRACTICAL RECOMMENDATIONS

We recommend reporting results under both protocols:

- **Random protocol** for compatibility with prior work and standard leaderboard settings;
- **EDGE protocol** to provide boundary-aware evaluation and a more informative characterization of performance variability across class orders.

This combined reporting provides a clearer picture of a method's robustness under realistic order distributions and mitigates the risk of drawing conclusions from a small number of random orders.

**Dataset support.** The current release provides built-in support for **CIFAR-100**, **CUB-200**, **ImageNet-R**, and **ImageNet-A**. To extend EDGE to additional datasets, users need to register the corresponding *class label names* (used to compute CLIP text-embedding similarities) in `utils/edge.py`. Concretely, add the dataset-specific list/dictionary of class names in the same format as the existing entries, and ensure the dataset identifier in your configuration matches the key used in `utils/edge.py`.

