# OpenReview forum: "The Lie of the Average: How Class Incremental Learning Evaluation Deceives You?"
_ICLR.cc/2026/Conference — ICLR 2026 Poster_

### Official Review · Reviewer_SxFj · 2025-10-30

**Soundness:** 3
**Presentation:** 3
**Contribution:** 3
**Rating:** 6
**Confidence:** 3

**Summary:**

The paper is about fixing misleading evaluation in class incremental learning (CIL). The authors propose EDGE, an extreme-case-based protocol that samples hardest, easiest, and random class sequences guided by inter-task similarity computed from CLIP text embeddings.

**Strengths:**

1. The paper reframes CIL evaluation from mean/variance over a few random orders to estimating the full performance distribution.
2. EDGE remains effective across datasets (CIFAR‑100, CUB‑200, ImageNet‑R), backbones (ResNet, ViT), and CLIP encoder sizes.
3. This paper is fluently written and easy to read.

**Weaknesses:**

1. Missing baselines.  RS is treated as a weak baseline (3 sequences with fixed seeds), while more principled sampling exists (stratified, Latin hypercube, coverage maximization, Bayesian optimization).

**Questions:**

In the EDGE protocol design section, it mentions that the extreme sequence generation algorithm uses CLIP text encoding to construct class similarity matrices. Could you provide more details on how this process works and how it contributes to generating high and low similarity sequences? Additionally, what are some potential limitations or challenges associated with using CLIP for this purpose?

---

> ### Author Response · Authors · 2025-11-19
> **Response (1/2)**
>
> Thanks for the valuable feedback. In this rebuttal, we address your concerns in a Q&A format:
>
>
> **W1:** Missing baselines
>
> **A1:** Thank you for the suggestion. The sequence space is extremely large. Some of the systematic sampling schemes you mentioned, such as stratified sampling and Latin hypercube sampling, are infeasible in our setting because their sample efficiency and computational cost are difficult to control in a high dimensional discrete sequence space. We have added **four additional baselines**:
>
> **(1) LLM-generated sequences (LLM-1, $\approx$130 s per sequence):** A single-round generation procedure in which a large language model directly produces a candidate sequence based on a prompt describing “easy” or “hard'' sequences. This baseline tests whether semantic difficulty can be inferred directly from class names without any iterative refinement.
>
> **(2) LLM-generated sequences (LLM-5, $\approx$600 s per sequence):** A five-round iterative refinement procedure. Each round, the LLM receives feedback regarding the previously generated sequence and attempts to correct or adjust its output in the next iteration. This baseline evaluates whether multi-step reasoning helps the LLM better capture difficulty.
>
> **(3) Adversarial Sampling (AS, $\approx$0.9 s per sequence):** A greedy, similarity-based adversarial strategy. At each step, AS selects the class that is maximally dissimilar from all currently selected classes, thereby increasing sequence difficulty by pushing the sequence toward the tail of the similarity distribution.
>
> **(4) Max-cover Sampling (MS, $\approx$8 s per sequence):** A randomized search-based approach. We first sample a pool of candidate sequences (we use 200), compute for each sequence a coverage or farthest-distance score relative to previously selected sets, and finally choose the top-ranked sequences.
>
> The results are summarized in the table below (detailed procedures and a more readable version of the table are provided in **Appendix E4**).
>
> | Method      | EDGE Hard | RS Hard | AS Hard | MS Hard | LLM-1 Hard | LLM-5 Hard |
> | ----------- | --------- | ------- | ------- | ------- | ---------- | ---------- |
> | L2p         | **58.35** | 62.58   | 61.75   | 62.60   | 64.97      | 67.01      |
> | CODA-Prompt | **65.65** | 67.42   | 65.96   | 66.13   | 67.01      | 65.78      |
> | Hide-Prompt | **80.52** | 81.45   | 81.02   | 80.96   | 82.35      | 80.89      |
> | RanPAC      | **88.68** | 88.72   | 88.99   | 88.72   | 88.72      | 88.72      |
> | EASE        | **84.29** | 84.60   | 84.69   | 84.39   | 84.56      | 84.82      |
> | MOS         | **87.69** | 88.49   | 88.76   | 88.38   | 89.13      | 88.23      |
>
> | Method      | EDGE Easy | RS Easy | AS Easy | MS Easy | LLM-1 Easy | LLM-5 Easy |
> | ----------- | --------- | ------- | ------- | ------- | ---------- | ---------- |
> | L2p         | **72.13** | 66.21   | 69.48   | 63.70   | 66.92      | 67.68      |
> | CODA-Prompt | 69.42     | 68.12   | 67.33   | 68.18   | 68.58      | **69.59**  |
> | Hide-Prompt | **83.21** | 82.35   | 82.39   | 82.63   | 82.36      | 81.45      |
> | RanPAC      | **89.40** | 89.15   | 89.21   | 89.21   | 89.25      | 88.68      |
> | EASE        | **85.37** | 84.96   | 85.07   | 85.33   | 85.01      | 84.78      |
> | MOS         | **89.56** | 88.98   | 88.93   | 88.30   | 88.56      | 89.26      |
>
> From these results, we can draw the following conclusions:
>
> (1) The LLM-based generation methods, while sometimes effective, are both **costly** and **unstable**. Single-round generation (LLM-1) exhibits high variance and frequently fails to produce consistently hard sequences. Iterative prompting (LLM-5) significantly improves stability and typically outperforms RS, yet in most cases it still falls far short of EDGE. More importantly, multi-round LLM workflows are **several orders of magnitude slower**, making them impractical for large-scale or low-latency evaluation.
>
> (2) Sampling-based methods can identify difficult sequences but lack **transferability**. AS and MS are computationally efficient and can discover orderings that challenge specific algorithms; however, their performance does **not generalize** across the diverse set of CIL methods we evaluate. In contrast, EDGE reliably identifies extreme sequences that remain difficult for a broad range of methods, striking a more desirable balance between **effectiveness** and **generality**.
>
> In summary, the additional baselines demonstrate that EDGE, informed by learning-theoretic insights and empirical observations, achieves clear advantages in both efficiency and robustness and is an effective method for generating extreme sequences.

---

> > ### Author Response · Authors · 2025-11-19
> > **Response (2/2)**
> >
> > **Q1:** Could you provide more details on how the process works and how it contributes to generating high and low similarity sequences
> >
> > **A2:** Please refer to **Appendix D.2** for full details. The appendix contains **pseudocode** for Algorithm 1 and Algorithm 2, **step by step analysis**, a **schematic diagram** in Figure A3, and **theoretical results** including optimality bounds and complexity analysis.  If you need any further information, we welcome further discussion.
> >
> > **Q2:** Additionally, what are some potential limitations or challenges associated with using CLIP for this purpose
> >
> > **A3:** The main limitation of CLIP is its limited direct transferability in specialized domains, particularly when class names are technical or abstract terms, which can reduce the alignment between CLIP text embeddings and image semantics. For example, in visual specialized domains, class names may be highly technical, and CLIP struggles to interpret and encode them. However, we provide a mitigation strategy in **Appendix E.3.2**, which involves expanding class names into visually descriptive captions. We validated this approach on **EyePACS** and **HAM10000**, where GPT-5-based expansions significantly improved the agreement between text-based and image-based similarities.

---

> > > ### Author Response · Authors · 2025-11-27
> > >
> > > Dear Reviewer SxFj,
> > >
> > > We hope this message finds you well. As **the discussion phase is approaching its final week**, we would like to kindly check whether our rebuttal has satisfactorily addressed your concerns. If there are any remaining questions or aspects that may benefit from further clarification, please feel free to let us know. We would be glad to provide additional explanations.
> > >
> > > Your feedback is very important to us, and we sincerely appreciate the time and effort you have devoted to reviewing our work.
> > >
> > > Best regards,
> > >
> > > Authors

---

### Official Review · Reviewer_Qscv · 2025-10-31

**Soundness:** 3
**Presentation:** 3
**Contribution:** 3
**Rating:** 8
**Confidence:** 5

**Summary:**

This paper critiques the evaluation protocol in Class Incremental Learning (CIL), which relies on averaging results over a few random class-order sequences. The authors theoretically and empirically demonstrate that such Random Sampling (RS) severely underestimates variance and produces biased means due to the factorial explosion of possible class sequences. To address this, they propose EDGE (Extreme case–based Distribution & Generalization Evaluation), a new evaluation protocol that identifies easy and hard cases using inter-task similarity derived from CLIP text embeddings. They claim that, evaluating models on these extreme and median sequences better approximates the true performance distribution.

**Strengths:**

1. Challenges a long-standing but under-scrutinized assumption in CIL benchmarking, introducing a valuable distributional perspective.
2. Provides clear theorems quantifying the inadequacy of random sampling.
3. Demonstrates consistency across model types (pre-trained and non-pre-trained) and multiple datasets.

**Weaknesses:**

1. Lack of real-world validation: Although the authors repeatedly motivate their work through real-world continual learning scenarios (e.g., autonomous driving, uncontrolled class emergence), all experiments are performed on simple disjoint data streams. They do not include any real-world or realistic task-order settings, such as those in Real-World Continual Learning[1] or I-Blurry[2]/Si-Blurry[3] setups, which explicitly simulate gradual or overlapping task transitions.
2. Dependence on CLIP embeddings: The approach assumes that CLIP’s text-space similarity correlates with inter-task difficulty, which may not hold in all domains (e.g., non-visual or specialized data).
3. Edge cases unexplored: The paper could discuss scenarios where inter-task similarity fails to predict performance variance (e.g., highly regularized models).
4. Overlooked fairness implications: Focusing on extreme sequences might overemphasize worst-case behavior, potentially penalizing stable models that perform consistently across moderate task orders.

citations:
[1]Online Class-Incremental Learning For Real-World Food Image Classification (Siddeshwar Raghavan, Jiangpeng He, Fengqing Zhu)
[2] Online Continual Learning on Class Incremental Blurry Task Configuration with Anytime Inference (Hyunseo Koh, Dahyun Kim, Jung-Woo Ha, Jonghyun Choi)
[3]Online Class Incremental Learning on Stochastic Blurry Task Boundary via Mask and Visual Prompt Tuning (Jun-Yeong Moon, Keon-Hee Park, Jung Uk Kim, Gyeong-Moon Park)

**Questions:**

1. How robust is the inter-task similarity heuristic across modalities or non-visual datasets? Could EDGE generalize beyond CLIP-based text embeddings?
2. For large-scale settings where enumerating even three sequences per dataset is costly, how would EDGE be adapted or approximated?
3. Does emphasizing extreme sequences risk over-penalizing models optimized for stability rather than adaptability?

I am willing to change my review if the concerns are addressed.

---

> ### Author Response · Authors · 2025-11-19
> **Response (1/2)**
>
> Thanks for the valuable feedback. In this rebuttal, we address your concerns in a Q&A format:
>
> **W1:** Lack of real world validation
>
> **A1** We acknowledge this concern. The classical class incremental learning setting assumes disjoint classes across tasks, which allows us to design class arrival orders to produce extreme sequences for evaluation. The Online CIL and I-Blurry settings you mentioned are important. In those settings some of the theory and assumptions in our paper may **need revision or extension**. We plan to study these scenarios in future work, including both theoretical adjustments and empirical validation.
>
> **W2, Q1:** Dependence on CLIP embeddings
>
> **A2:** EDGE relies on CLIP embeddings because our setup assumes downstream task data are not directly available. If downstream data are available, one can compute inter-task similarity directly from those images and avoid using CLIP.
>
> To evaluate feasibility in specialized domains, we conducted experiments on two medical imaging datasets: **EyePACS** [1] and **HAM10000** [2]. For each dataset, we compared the inter-class similarity matrix computed from **CLIP text embeddings of class names** with that computed from **image-derived prototype representations**. Their consistency was measured using Spearman rank correlation. Detailed procedures are provided in **Appendix E.3.2**.
>
> The results are summarized below:
>
> | Dataset  | Text from class names $\rho (p)$ | Text after LLM-based augmentation $\rho (p)$ |
> | :------: | :------------------------------: | :------------------------------------------: |
> | EyePACS  |           0.588 (0.07)           |                 0.863 (0.01)                 |
> | HAM10000 |           0.279 (0.22)           |                 0.653 (0.02)                 |
>
> The first column shows that directly using raw class names leads to weak alignment between text- and image-based similarities, confirming the reviewer’s concern. However, we find that this issue can be largely resolved with **a simple text-augmentation step** using a generative language model. After applying this LLM-based augmentation (see Appendix E.3.2), the alignment is substantially **improved** on both datasets.
>
> This indicates that, although the initial CLIP text embeddings may be suboptimal in such specialized domains, the issue is easy to remedy in practice and should not be a major concern.
>
> In non visual domains it is equally important to **find suitable proxies that reflect inter-task similarity**. Once such proxies are available, EDGE can be applied in the same spirit.
>
> [1] Kaggle-EyePACS dataset
>
> [2] HAM10000: A large collection of dermatoscopic images of common pigmented skin lesions

---

> > ### Author Response · Authors · 2025-11-19
> > **Response (2/2)**
> >
> > **W3:** Edge cases unexplored
> >
> > **A3:** In our validation experiments (Figure 2c) and theoretical analysis (Theorem 3), task difficulty is shown to correlate with inter-task similarity. This allows us to use task similarity to **estimate the model’s eventual performance**. The edge cases you mention are certainly interesting; however, we have not yet identified concrete scenarios where they occur.
> >
> > **W4, Q3:** Overlooked fairness implications
> >
> > **A4:** Rest assured, EDGE does not undermine fairness. EDGE offers a **more diverse set of evaluation perspectives** and thus supports fairer choices. In practice EDGE reports multiple metrics such as lower bound, upper bound and variance. If an application prioritizes a higher lower bound, one can choose the method with the highest lower bound. If stability is the priority, one can choose the method with the smallest variance. These evaluation dimensions are complementary and **help decision makers select a method according to real needs.**
> >
> > **Q2:** For large scale settings where enumerating even three sequences per dataset is costly, how would EDGE be adapted or approximated?
> >
> > **A5:** Your concern is reasonable, but we believe that local and controlled evaluation before deployment is essential. **Any machine learning system intended for deployment should undergo local verification to ensure safety, and some evaluation cost is unavoidable.**
> >
> > Importantly, EDGE **does not require excessive sequence generation time.** For example, splitting 200 classes into 10 tasks and running EDGE sequence generation takes about **0.9 seconds** on a standard CPU. Therefore in most practical verification settings producing a small number of representative extreme sequences with EDGE is time efficient and feasible.

---

> > > ### Author Response · Authors · 2025-11-27
> > >
> > > Dear Reviewer Qscv,
> > >
> > > We hope this message finds you well. As **the discussion phase is approaching its final week**, we would like to kindly check whether our rebuttal has satisfactorily addressed your concerns. If there are any remaining questions or aspects that may benefit from further clarification, please feel free to let us know. We would be glad to provide additional explanations.
> > >
> > > Your feedback is very important to us, and we sincerely appreciate the time and effort you have devoted to reviewing our work.
> > >
> > > Best regards,
> > >
> > > Authors

---

### Official Review · Reviewer_rrMk · 2025-10-31

**Soundness:** 3
**Presentation:** 3
**Contribution:** 2
**Rating:** 4
**Confidence:** 3

**Summary:**

This paper critiques the current evaluation methodology used in class-incremental learning (CIL), where a small number of randomly sampled class orders are used to estimate model performance. The authors argue that this common practice does not accurately approximate the true expected performance across all possible class sequences. To address this, the paper proposes an improved evaluation approach designed to more reliably estimate true model performance.

**Strengths:**

- This paper investigates an evaluation issue in CIL that is often overlooked in the literature, which is an interesting direction for the CL community.

- The paper is clearly written and well structured, with both theoretical and empirical analysis supporting the proposed evaluation method.

- Highlighting the impact of class ordering contributes valuable insight toward more trustworthy benchmarking in CIL.

**Weaknesses:**

- The motivation of the paper raises concerns. The introduction uses real-world scenarios (e.g., self-driving cars) where the learner does not have control over class arrival order. However, the proposed method assumes class ordering can be controlled or enumerated for evaluation, which conflicts with the motivation.
- Although the common RS (Random Sampling) evaluation does not estimate the true model accuracy, it does enable fair comparison among methods if all algorithms are evaluated using the same sampled sequences. Since CIL benchmarks are used for comparison of CIL algorithms rather than real-world deployment, the evaluation of existing literature is not entirely unreasonable.
- A key practical question would be whether a method that performs well on difficult sequences also performs well on easy sequences. A more actionable direction may be benchmarking both cases and identifying algorithms that consistently perform well regardless of class order.
- The existence of hard vs. easy class sequences is already known in CIL. Thus, the novelty of the paper is mainly limited to proposing a new evaluation metric. Without demonstrating an algorithm that improves under the new metric, the contribution may feel incomplete.
- The experiments show limited benefit in the practical application of ranking existing CL algorithms (referring to Table 2).
  - Example: CIFAR-100 (hard sequence): both True and RS rankings identify iCaRL as best. Here, EDGE does not contribute any new information in terms of ranking existing CL algorithms
  - Example: ImageNet-R (easy sequence): True says iCaRL as best, but RS and EDGE disagree. Here, neither estimator consistently reflects the true ranking.

**Questions:**

- For experiments using pre-trained models, what dataset is used to pre-train the backbone?
- Why are RS and EDGE results very close in Table 1? Does this imply that EDGE only offers minor improvements over RS in most realistic cases?
- Can methods that perform well on difficult sequences also automatically perform well on easy sequences? If not, what may be the reason behind it?

---

> ### Author Response · Authors · 2025-11-19
> **Response (1/2)**
>
> Thanks for the valuable feedback. In this rebuttal, we address your concerns in a Q&A format:
>
> **W1:** The motivation of the paper raised concerns.
>
> **A1:** We do not view these points as contradictory. We believe that before a CIL algorithm is deployed, it should undergo two stages:
>
> (1) **Evaluation on controllable sequences**, where artificially constructed sequences of varying difficulty are used to test the model’s robustness and assess potential risks.
>
> (2) **Deployment of the best-performing model** (selected based on stage (1)) to real-world scenarios, where task ordering is inherently uncontrollable.
>
> **The first stage is essential.** Skipping it and deploying a model directly may lead to safety risks and financial losses.
>
> Our work focuses on the first stage and aims to improve method selection through more stringent evaluations on sequence difficulty, thereby **enhancing the safety and reliability of the final system**.
>
> **W2:** Fairness of comparison and actual deployment
>
> **A2:** We believe that fairness cannot be defined in such a simplistic manner. The reason you mentioned, that "evaluation does not estimate the true model accuracy," is precisely what leads to unfairness. Consider a simple scenario: a given random sequence could be in the top five percent for L2P but in the bottom five percent for MOS, which is clearly disadvantageous to MOS. Similar phenomena are observed in **Figure 1, Figures 2a and 2b, and Figure 5**. EDGE aims to **identify sequences that are broadly hard or broadly easy across methods**, and such sequences provide more representative and fairer evaluations.
>
> Applying this improved evaluation to deployment is both reasonable and necessary, but this step is not yet fully mature. The continued use of RS in current CIL benchmarks represents a **long-standing issue** that the community should resolve before moving toward deployment-focused evaluation protocols.
>
> **W3, Q3:** A key practical question would be whether a method that performs well on difficult sequences also performs well on easy sequences. A more actionable direction may be benchmarking both cases and identifying algorithms that consistently perform well regardless of class order.
>
> **A3:** Clearly, this is not always the case. In Figure 5(a), CODA-Prompt and Hide-Prompt provide counterexamples, and in Figure 5(b), Hide-Prompt and EASE do so as well. EDGE is designed to **benchmark both difficult and easy sequences**, providing a more comprehensive evaluation.
>
> This inconsistency also offers insights for method selection based on application needs: if the goal is to maximize the lower bound of performance, one might choose CODA-Prompt or Hide-Prompt; if the goal is to maximize the upper bound, Hide-Prompt or EASE would be more suitable.
>
> **W4:** Contribution of the paper
>
> **A4:** We would like to reiterate the novelty and contributions of our work from the following three perspectives:
>
> (1) We agree that the existence of hard versus easy class sequences in CIL has been observed before. **However**, prior work has **rarely** examined how this phenomenon affects the evaluation protocol. The common RS evaluation protocol fails to sample those representative hard or easy sequences, which **introduces bias into evaluation**. Our paper focuses on exposing and quantifying this evaluation bias so the community can properly account for it.
>
> (2) We disagree with the characterization that our novelty is mainly limited to proposing a new metric. First, we **provide substantial theoretical analysis** showing why RS can produce misleading evaluations and under what conditions this occurs. Second, we **do not** merely propose a single metric; we **introduce a new evaluation protocol** (EDGE) that actively generates and captures extreme sequences to produce a more comprehensive and robust evaluation. **This is a methodological contribution beyond a simple metric.**
>
> (3) We acknowledge that designing methods guided by EDGE is a meaningful next step and we discuss this direction in the main text. Nevertheless, the primary objective of this paper is to **raise awareness of the limitations of RS-based evaluation** and to provide a more appropriate evaluation protocol. **Encouraging the community to improve evaluation practices is itself a substantial contribution.**

---

> ### Author Response · Authors · 2025-11-19
> **Response (2/2)**
>
> **W5:** The experiments show limited practical benefit in ranking existing CL algorithms (as shown in Table 2).
>
> **A5:** We would like to emphasize that EDGE is not designed to claim that RS is entirely wrong. Instead, **in most cases**, EDGE outperforms RS, and therefore it is natural that the rankings from the two protocols coincide in some scenarios.
>
> For detailed ranking differences, we refer you to **Appendix E.1.3**. On CIFAR-100, **EDGE exactly matches the ground-truth ranking** (distance = 0), whereas RS deviates significantly (distance = 3.5). On ImageNet-R, although EDGE does not perfectly match the true ranking, **it still substantially outperforms RS** (distance = 4.4 vs. 8.1).
>
> Evaluation methodology has long been under-explored in the CIL community. **EDGE is our attempt at addressing this important issue, and we believe future efforts should involve broader community collaboration to further develop benchmarking standards.**
>
> **Q1:** For experiments using pre-trained models, what dataset is used to pre-train the backbone?
>
> **A6:** Following the settings of [1, 2], we use ViT-B/16 pre-trained on ImageNet-21K as the backbone for all pre-trained methods to **ensure a fair comparison.**
>
> [1] Pilot:A pre-trained model-based continual learning toolbox.
>
> [2] Ranpac:Random projections and pre-trained models for continual learning.
>
> **Q2:** Why are the RS and EDGE results very close in Table 1? Does this imply that EDGE only provides minor improvements over RS in most realistic cases?
>
> **A7:** In Table 1, for methods with a large performance distribution range, such as L2P and CODA-Prompt, **EDGE captures much more extreme values compared to RS**, because the available **improvement space is large**. RS and EDGE appear similar only for methods such as RanPAC, EASE, and MOS, which are relatively robust to task ordering. As discussed in **detail in Lines 424–429 of the main text**, these methods have a small potential sequence space and **narrow performance ranges**, which makes the results appear close.
>
> However, in practical terms, they **should not be considered truly close.** Computing the absolute difference between EDGE and RS is not meaningful; instead, one should consider the improvement relative to the available performance range.
>
> For example, on CIFAR-100 with EASE, EDGE fully captures the lower bound (95.33). Although RS is only 0.17 higher, this is simply because the improvement space is only 0.17. Most other seemingly “close” cases follow the same reasoning. Therefore, this does **not imply that EDGE only offers minor improvements over RS in most realistic cases**.

---

### Official Review · Reviewer_6DZz · 2025-11-01

**Soundness:** 2
**Presentation:** 3
**Contribution:** 2
**Rating:** 4
**Confidence:** 4

**Summary:**

This paper focuses on the evaluation protocol of continual learning. Under the standard evaluation protocol, Random Sampling (RS) tends to be unfair, as it emphasizes the mean rather than the deviation. Consequently, the EDGE protocol is proposed for improved sampling of task sequences. First, the author identifies that the difficulty of the continual learning (CL) sequence is highly correlated with inter-task similarity. Therefore, EDGE samples "easy" and "hard" sequences based on the similarity calculated using the CLIP model. Experimental results show that EDGE provides a better estimation of the performance range of CL methods.

**Strengths:**

1. The proposed relationship between task similarity and difficulty is intuitive and interesting.
2. Under the existing setup (3 runs for each method), the proposed EDGE can be much stronger compared with RS.
3. The source code is partially included in the submission.

**Weaknesses:**

1. It is true that recent continual learning (CL) papers consistently use Random Sampling (RS) in their evaluations. However, the experimental setting of this manuscript assumes that only three runs are used for the evaluation, which does not fully simulate a realistic Class Incremental Learning (CIL) evaluation protocol. According to basic statistical theory, the estimation provided by the RS protocol will be much closer to reality if more runs are employed. As this paper focuses on the fairness of the evaluation protocol, it is essential to consider scenarios where the number of runs exceeds three for a factual and equitable comparison, and to examine how performance differences scale with an increased number of runs.
2. EDGE fails to account for cases where analytic continual learning methods are evaluated, such as GACL [A] and RanPAC. The performance of these methods is usually minimally impacted by task order. This may explain the similarity in RS evaluation for RanPAC and the EDGE evaluation, despite the fact that RanPAC fine-tunes the backbone using the training samples.
3. As an evaluation protocol, EDGE only takes final accuracy into consideration. However, it is well known that final accuracy is not the sole important factor. To evaluate the dynamics of the CL learner, metrics such as anytime inference [B] and plasticity/forgetting metrics are also vital, yet these are unfortunately not discussed within the proposed metrics.
4. Methods such as ConvPrompt [C] and SEMA [D] take inter-task similarity into account. Including such methods in the benchmarking would be beneficial.

References:

    [A] Zhuang, Huiping, et al. "GACL: Exemplar-free generalized analytic continual learning." Advances in Neural Information Processing Systems 37 (2024): 83024-83047.
    [B] Koh, Hyunseo, et al. "Online continual learning on class incremental blurry task configuration with anytime inference." arXiv preprint arXiv:2110.10031 (2021).
    [C] Roy, Anurag, et al. "Convolutional prompting meets language models for continual learning." Proceedings of the IEEE/CVF Conference on Computer Vision and Pattern Recognition. 2024.
    [D] Wang, Huiyi, et al. "Self-expansion of pre-trained models with mixture of adapters for continual learning." Proceedings of the Computer Vision and Pattern Recognition Conference. 2025.

**Questions:**

For the paper to be deemed acceptable, I believe the following question is crucial:
1. How do Random Sampling (RS) and EDGE perform with an increased number of sampled runs?

---

> ### Author Response · Authors · 2025-11-19
> **Response (1/2)**
>
> Thanks for the valuable feedback. In this rebuttal, we address your concerns in a Q&A format:
>
> **W1, Q1:** Multiple runs experiment (core concern you raised)
>
> **A1:** In the main text we followed the evaluation practice used by most prior works and reported results from three runs. We agree that additional verification over multiple runs is important. We **conducted two experiments** to study how RS behaves as the number of samples increases and to **further demonstrate EDGE’s advantages**.
>
> **\[Experiment 1]**: Enumerable experiments (corresponding to Section 5.1)
>
> We visualized **how evaluation results change as the number of RS samples increases** when evaluating pretrained methods (see Figure A6 in Appendix E.5). From these results we draw the following conclusions:
>
> (1) In most settings, RS reaches the distribution estimation quality of EDGE with 3 EDGE samples after **roughly 5 to 6 RS samples**. In other words, on an evaluation set with only 90 sequences, RS typically requires about twice the evaluation time to achieve distribution estimation quality comparable to EDGE.
>
> (2) If we also increase the number of EDGE samples, EDGE remains superior to RS. After about **10 to 20 samples**, the distribution estimation capabilities of the two methods **converge**.
>
> **\[Experiment 2]**: Classic CIL settings (corresponding to Section 5.2)
>
> For time reasons we used the CUB dataset and selected L2P (wide performance range) and EASE (narrow performance range) as representative cases. We repeatedly performed random sampling and evaluation to determine **how many RS samples are needed to capture the extreme sequences that EDGE can capture in a single sampling**. We visualized the sampling distributions and marked EDGE’s positions (see Figure A7 in Appendix E.5). From these results we draw the following conclusions:
>
> From these results we draw the following conclusions:
>
> (1) For L2P, under our setup EDGE found a hard-case accuracy of **58.5** and an easy-case accuracy of **72.3.** We ran **600 RS samplings**, of which **only 6** found lower accuracies **(minimum 57.55)**, and **none** found a higher accuracy than EDGE.
>
> (2) For EASE, under our setup EDGE found a hard-case accuracy of **84.29** and an easy-case accuracy of **85.37.** After more than **400** random evaluations, **only 4** RS samples found lower accuracies **(minimum 84.09)**, and **only 1** sample found a higher accuracy (**85.41**).
>
> **Conclusion**
> These results confirm the intuitive fact that increasing RS sample count improves RS. However, such improvement typically entails a large evaluation cost. Compared with EDGE, discovering extreme sequences using only RS is both time consuming and inefficient.

---

> > ### Author Response · Authors · 2025-11-19
> > **Response (2/2)**
> >
> > **W2:** Impact of EDGE on analytic continual learning methods
> >
> > **A2:** We agree that analytic continual learning methods are, by design, more robust to task ordering. EDGE explicitly accounts for this situation. Take **RanPAC** as an illustrative example: RanPAC’s final performance depends primarily on which classes are used to fine-tune the backbone in the **first task**. According to **Lemma 1**, this means that its effective sequence space is extremely small. Consequently, RS can often cover this reduced space reasonably well.
> >
> > However, **EDGE can still identify meaningful extreme cases**, even for such analytic methods. This is because EDGE determines the **first task** using *global task similarity*:
> >
> > - for generating **hard** sequences, EDGE selects the task with the **lowest global similarity** to all others;
> > - for generating **easy** sequences, EDGE selects the task with the **highest global similarity**.
> >
> > The subsequent steps follow the procedures detailed in **Algorithm 1** and **Algorithm 2** in **Appendix D.2**.
> >
> > As show in **Table1 , Figure 5, and Table A7**, we observe that sequences generated through this global-similarity-based initialization do lead to performance that is noticeably better or worse than what RS produces. This demonstrates that even for analytic CIL methods with limited sequence sensitivity, EDGE remains capable of revealing extreme cases that RS may miss.
> >
> > **W3:** Other evaluation metrics
> >
> > **A3:** Indeed, other metrics such as forgetting and anytime inference are also important for continual learning. However, both metrics are fundamentally **derived from accuracy**: anytime inference computes accuracy at every timestep and summarizes it via AUC, while forgetting is typically defined as the difference between the final accuracy and the accuracy at the time each class was first learned. Thus, these metrics ultimately rely on accuracy evaluations.
> >
> > This paper focuses on addressing a long-standing issue in continual learning evaluation. To avoid introducing unnecessary complexity, we restrict our study to accuracy-based evaluation. Nevertheless, it is important to note that our methodology and analytical framework can be **readily extended to other metrics**, making them suitable for future work or expanded empirical studies.
> >
> > **W4:** Supplement: ConvPrompt and SEMA
> >
> > **A4**: Thank you for the suggestion. Unfortunately, ConvPrompt does not currently support arbitrary, user-specified evaluation orders, so we are unable to provide ConvPrompt results under the requested setting in this response. We will implement this functionality as soon as possible.
> >
> > The results for SEMA under the requested evaluation protocol are reported in the table below.
> >
> > |      Metric      |  Min  |  Max  | Mean  | Std  |
> > | :--------------: | :---: | :---: | :---: | :--: |
> > |   CIFAR100, RS   | 86.07 | 86.77 | 86.47 | 0.29 |
> > |  CIFAR100, EDGE  | 85.10 | 87.95 | 86.37 | 1.18 |
> > |    CUB200, RS    | 71.54 | 72.31 | 71.91 | 0.31 |
> > |   CUB200, EDGE   | 67.22 | 80.96 | 73.49 | 5.67 |
> > |  ImageNet-R, RS  | 74.13 | 74.75 | 74.50 | 0.26 |
> > | ImageNet-R, EDGE | 73.38 | 75.95 | 74.53 | 1.06 |
> >
> > These results indicate that EDGE still **captures more extreme sequences**. After we correct ConvPrompt’s implementation, we will add the full comparison and update the table and analysis in the final manuscript.

---

> > > ### Comment · Reviewer_6DZz · 2025-11-26
> > >
> > > I would like to thank the authors for their effort in the rebuttal. After checking the answers, my concerns are addressed. Thus, after also considering the reviews from other reviewers, I would calibrate my score to 6.

---

> > > > ### Author Response · Authors · 2025-11-26
> > > > **Thank you for your positive feedback**
> > > >
> > > > Dear Reviewer 6DZz,
> > > >
> > > > We are glad that our rebuttal addressed most of your concerns and appreciate that you increased your rating to 6. Your suggestions were very helpful and have materially improved the paper. If you have any remaining comments or would like further clarification on any point, we would be happy to respond.
> > > >
> > > > Best regards,
> > > >
> > > > Authors

---

### Official Review · Reviewer_Htje · 2025-11-01

**Soundness:** 3
**Presentation:** 3
**Contribution:** 3
**Rating:** 6
**Confidence:** 4

**Summary:**

This paper discusses the evaluation protocol of the class incremental learning and shows that the widely used random sampling protocol fails to capture the full distribution of model performance. To solve this problem, the authors propose an evaluation framework, EDGE, that finds extreme class sequences based on inter-task similarity computed via CLIP text embeddings. Both theoretical analysis and empirical experiments demonstrate that EDGE offers a more accurate estimation of performance distribution on CIFAR-100, CUB-200, and ImageNet-R, EDGE better captures upper/lower bounds and reduces distribution distance versus RS, remaining robust across different model backbones and CLIP encoder sizes.

**Strengths:**

- This paper addresses the limitations of evaluation protocols in CIL, a critical area gaining increasing attention.
- The theoretical analysis is clear and robust, highlighting the fundamental limitations of random sampling (RS) protocols.
- The proposed method is straightforward and easy to implement. Utilizing CLIP-based label embeddings to evaluate inter-task similarity appears appropriate for most tasks.
- The comparison of RS and EDGE is analyzed from various perspectives, emphasizing the limitations of random sampling.
- Both non-pretrained and pretrained CIL methods were assessed using two evaluation protocols (random sampling & EDGE), offering valuable insights for researchers.

**Weaknesses:**

- Using CLIP text embeddings for class similarity in non-natural image datasets, like medical imaging or remote sensing, may lead to inaccuracies due to potential misalignment between label semantics and visual similarity.

- The comparison relies solely on random sampling, without addressing other potential baseline evaluation methods.

- The assumption that inter-task semantic similarity perfectly correlates with task difficulty may not be universally valid, particularly in areas where semantic similarity fails to encompass the full complexity of tasks.

- The application scope of this evaluation protocol requires further discussion. Specifically, how crucial is it to capture performance distribution and absolute accuracy compared to solely focusing on relative model ranking? Is this type of evaluation necessary for all continual learning scenarios, or is it primarily relevant for certain deployment-critical applications?

**Questions:**

1) The paper emphasizes that RS leads to inaccurate variance estimation and poor coverage of extreme cases, and that EDGE improves the estimation of the performance distribution. While this is well motivated, it raises a fundamental question about what the community should prioritize in continual learning evaluation. In many continual learning scenarios, our focus may be more on relative model ranking rather than on absolute accuracy metrics.

2) The proposed framework relies on the assumption that semantic similarity between class labels, derived from CLIP text embeddings, correlates with inter-task similarity and reflects task difficulty. How robust is this assumption beyond natural image datasets? In domains such as medical imaging or synthetic datasets, label names may inadequately represent visual or distributional similarities between tasks.

3) The framework may be vulnerable to data contamination, especially since both the evaluated continual learning (CL) methods and the CLIP model used for task similarity computation are pre-trained on extensive web-scale datasets. Have the authors considered how data contamination—resulting from either model pre-training or benchmark overlap with CLIP—could affect task similarity estimates and, in turn, impact the reliability of the EDGE protocol?

4) When evaluating continual learning methods that are pre-trained (e.g., on CLIP or similar datasets), it's important to note that if the downstream datasets overlap with the pre-training corpus, task difficulty may be misrepresented. Tasks with greater overlap might seem "easier" not due to actual similarity, but because of data leakage from the pre-training phase.

5) How does RS improve with larger sample sizes? Specifically, how many RS sequences are required for RS to match EDGE in distribution approximation or ranking stability?

---

> ### Author Response · Authors · 2025-11-19
> **Response (1/4)**
>
> Thanks for the valuable feedback. In this rebuttal, we address your concerns in a Q&A format:
>
> **W1, Q2: Robustness of CLIP on non-natural images**
>
> **A1**: We agree with the reviewer’s concern regarding whether CLIP text embeddings remain reliable in non-natural image domains. To evaluate this, we conducted experiments on two medical imaging datasets: **EyePACS** [1] and **HAM10000** [2]. For each dataset, we compared the inter-class similarity matrix computed from **CLIP text embeddings of class names** with that computed from **image-derived prototype representations**. Their consistency was measured using Spearman rank correlation. Detailed procedures are provided in **Appendix E.3.2**.
>
> The results are summarized below:
>
> | Dataset  | Text from class names $\rho (p)$ | Text after LLM-based augmentation $\rho (p)$ |
> | :------: | :------------------------------: | :------------------------------------------: |
> | EyePACS  |           0.588 (0.07)           |                 0.863 (0.01)                 |
> | HAM10000 |           0.279 (0.22)           |                 0.653 (0.02)                 |
>
> The first column shows that directly using raw class names leads to weak alignment between text- and image-based similarities, confirming the reviewer’s concern. However, we find that this issue can be largely resolved with **a simple text-augmentation step** using a generative language model. After applying this LLM-based augmentation (see Appendix E.3.2), the alignment is substantially **improved** on both datasets.
>
> This indicates that, although the initial CLIP text embeddings may be suboptimal in such specialized domains, the issue is easy to remedy in practice and should not be a major concern.
>
> [1] Kaggle-EyePACS dataset
>
> [2] HAM10000: A large collection of dermatoscopic images of common pigmented skin lesions

---

> > ### Author Response · Authors · 2025-11-19
> > **Response (2/4)**
> >
> > **W2: Additional baselines**
> >
> > **A2:** Thank you for pointing this out. We have added **four additional baselines**:
> >
> > **(1) LLM-generated sequences (LLM-1, $\approx$130 s per sequence):** A single-round generation procedure in which a large language model directly produces a candidate sequence based on a prompt describing “easy” or “hard'' sequences. This baseline tests whether semantic difficulty can be inferred directly from class names without any iterative refinement.
> >
> > **(2) LLM-generated sequences (LLM-5, $\approx$600 s per sequence):** A five-round iterative refinement procedure. Each round, the LLM receives feedback regarding the previously generated sequence and attempts to correct or adjust its output in the next iteration. This baseline evaluates whether multi-step reasoning helps the LLM better capture difficulty.
> >
> > **(3) Adversarial Sampling (AS, $\approx$0.9 s per sequence):** A greedy, similarity-based adversarial strategy. At each step, AS selects the class that is maximally dissimilar from all currently selected classes, thereby increasing sequence difficulty by pushing the sequence toward the tail of the similarity distribution.
> >
> > **(4) Max-cover Sampling (MS, $\approx$8 s per sequence):** A randomized search-based approach. We first sample a pool of candidate sequences (we use 200), compute for each sequence a coverage or farthest-distance score relative to previously selected sets, and finally choose the top-ranked sequences.
> >
> > The results are summarized in the table below (detailed procedures and a more readable version of the table are provided in **Appendix E4**).
> >
> > | Method      | EDGE Hard | RS Hard | AS Hard | MS Hard | LLM-1 Hard | LLM-5 Hard |
> > | ----------- | --------- | ------- | ------- | ------- | ---------- | ---------- |
> > | L2p         | **58.35** | 62.58   | 61.75   | 62.60   | 64.97      | 67.01      |
> > | CODA-Prompt | **65.65** | 67.42   | 65.96   | 66.13   | 67.01      | 65.78      |
> > | Hide-Prompt | **80.52** | 81.45   | 81.02   | 80.96   | 82.35      | 80.89      |
> > | RanPAC      | **88.68** | 88.72   | 88.99   | 88.72   | 88.72      | 88.72      |
> > | EASE        | **84.29** | 84.60   | 84.69   | 84.39   | 84.56      | 84.82      |
> > | MOS         | **87.69** | 88.49   | 88.76   | 88.38   | 89.13      | 88.23      |
> >
> > | Method      | EDGE Easy | RS Easy | AS Easy | MS Easy | LLM-1 Easy | LLM-5 Easy |
> > | ----------- | --------- | ------- | ------- | ------- | ---------- | ---------- |
> > | L2p         | **72.13** | 66.21   | 69.48   | 63.70   | 66.92      | 67.68      |
> > | CODA-Prompt | 69.42     | 68.12   | 67.33   | 68.18   | 68.58      | **69.59**  |
> > | Hide-Prompt | **83.21** | 82.35   | 82.39   | 82.63   | 82.36      | 81.45      |
> > | RanPAC      | **89.40** | 89.15   | 89.21   | 89.21   | 89.25      | 88.68      |
> > | EASE        | **85.37** | 84.96   | 85.07   | 85.33   | 85.01      | 84.78      |
> > | MOS         | **89.56** | 88.98   | 88.93   | 88.30   | 88.56      | 89.26      |
> >
> > From these results, we can draw the following conclusions:
> >
> > (1) The LLM-based generation methods, while sometimes effective, are both **costly** and **unstable**. Single-round generation (LLM-1) exhibits high variance and frequently fails to produce consistently hard sequences. Iterative prompting (LLM-5) significantly improves stability and typically outperforms RS, yet in most cases it still falls far short of EDGE. More importantly, multi-round LLM workflows are **several orders of magnitude slower**, making them impractical for large-scale or low-latency evaluation.
> >
> > (2) Sampling-based methods can identify difficult sequences but lack **transferability**. AS and MS are computationally efficient and can discover orderings that challenge specific algorithms; however, their performance does **not generalize** across the diverse set of CIL methods we evaluate. In contrast, EDGE reliably identifies extreme sequences that remain difficult for a broad range of methods, striking a more desirable balance between **effectiveness** and **generality**.
> >
> > In summary, the additional baselines demonstrate that EDGE, informed by learning-theoretic insights and empirical observations, achieves clear advantages in both efficiency and robustness and is an effective method for generating extreme sequences.

---

> > > ### Author Response · Authors · 2025-11-19
> > > **Response (3/4)**
> > >
> > > **W3: Relationship between task similarity and task difficulty**
> > >
> > > **A3:** We **do not claim** that task difficulty is fully determined by inter-task similarity. Rather, we argue that task difficulty arises from multiple factors, most of which are difficult to quantify. Inter-task similarity is one of the few **quantifiable and analyzable** components, and, as shown in Fig. 2c, it exhibits a clear positive correlation with task difficulty. Therefore, our analysis approximates task difficulty **through the lens of inter-task similarity**, focusing on the aspect that can be measured and modeled in a principled manner.
> > >
> > > **W4,Q1: Application scenarios**
> > >
> > > **A4:** In applications where worst-case performance, robustness, and deployment safety are critical (e.g., autonomous driving, medical diagnosis), absolute accuracy and distributional behavior are crucial. EDGE is designed as a low-cost diagnostic tool that exposes extreme sequences that standard RS frequently fails to capture. **We do not claim that EDGE replaces RS; rather, it serves as a strong and practical complement to RS by revealing systematic behaviors that RS-based evaluation alone cannot uncover.**
> > >
> > > Moreover, even for model ranking, RS can mislead decisions. **Appendix E.1.3** presents a concrete case where EDGE guides model ranking: because RS is biased when estimating extreme sequences and the variance of model outputs, metrics derived from RS can produce misleading rankings; **EDGE more stably captures extreme sequences and yields rankings that are closer to the true ordering.**
> > >
> > > **Q3:** Effect of CLIP pretraining
> > >
> > > **A5:** We believe the effect is small. Prior work [3,4] shows that CLIP trained after pruning or filtering portions of LAION still retains strong generalization and zero-shot performance, indicating that **CLIP’s abilities are not solely driven by exact matches in the training set.**
> > >
> > > [3] Does CLIP’s generalization performance mainly stem from high train-test similarity?
> > >
> > > [4] A Closer Look at the Robustness of Contrastive Language-Image Pre-Training (CLIP)
> > >
> > > **Q4:** Effect of pretraining in CIL methods
> > >
> > > **A6:** You do not need to worry, **the pretrained model does not interfere with or bias our evaluation**. This is because:
> > >
> > > (1) the sequences generated by CLIP and our evaluation procedure are **decoupled** from how downstream methods are pretrained.
> > >
> > > (2) when comparing pretrained methods, **all methods use the same pretrained model and the same sequences**, so the difficulty is comparable and the comparison is fair.
> > >
> > > The issues you raise appear to pertain to the design of pretraining methods, rather than to the evaluation methodology we study.

---

> > > > ### Author Response · Authors · 2025-11-19
> > > > **Response (4/4)**
> > > >
> > > > **Q5:** Changes in RS as sampling count increases
> > > >
> > > > **A7:** Thank you for the suggestion. This analysis was necessary. We conducted two experiments to study how RS behaves as the number of samples increases and to further demonstrate EDGE’s advantages.
> > > >
> > > > **\[Experiment 1]**: Enumerable experiments (corresponding to Section 5.1)
> > > >
> > > > We visualized **how evaluation results change as the number of RS samples increases** when evaluating pretrained methods (see Figure A6 in Appendix E.5). From these results we draw the following conclusions:
> > > >
> > > > (1) In most settings, RS reaches the distribution estimation quality of EDGE with 3 EDGE samples after **roughly 5 to 6 RS samples**. In other words, on an evaluation set with only 90 sequences, RS typically requires about twice the evaluation time to achieve distribution estimation quality comparable to EDGE.
> > > >
> > > > (2) If we also increase the number of EDGE samples, EDGE remains superior to RS. After about **10 to 20 samples**, the distribution estimation capabilities of the two methods **converge**.
> > > >
> > > > **\[Experiment 2]**: Classic CIL settings (corresponding to Section 5.2)
> > > >
> > > > For time reasons we used the CUB dataset and selected L2P (wide performance range) and EASE (narrow performance range) as representative cases. We repeatedly performed random sampling and evaluation to determine **how many RS samples are needed to capture the extreme sequences that EDGE can capture in a single sampling**. We visualized the sampling distributions and marked EDGE’s positions (see Figure A7 in Appendix E.5). From these results we draw the following conclusions:
> > > >
> > > > From these results we draw the following conclusions:
> > > >
> > > > (1) For L2P, under our setup EDGE found a hard-case accuracy of **58.5** and an easy-case accuracy of **72.3.** We ran **600 RS samplings**, of which **only 6** found lower accuracies **(minimum 57.55)**, and **none** found a higher accuracy than EDGE.
> > > >
> > > > (2) For EASE, under our setup EDGE found a hard-case accuracy of **84.29** and an easy-case accuracy of **85.37.** After more than **400** random evaluations, **only 4** RS samples found lower accuracies **(minimum 84.09)**, and **only 1** sample found a higher accuracy (**85.41**).
> > > >
> > > > **Conclusion**
> > > > These results confirm the intuitive fact that increasing RS sample count improves RS. However, such improvement typically entails a large evaluation cost. Compared with EDGE, discovering extreme sequences using only RS is both time consuming and inefficient.

---

> ### Author Response · Authors · 2025-11-27
>
> Dear Reviewer Htje,
>
> We hope this message finds you well. As **the discussion phase is approaching its final week**, we would like to kindly check whether our rebuttal has satisfactorily addressed your concerns. If there are any remaining questions or aspects that may benefit from further clarification, please feel free to let us know. We would be glad to provide additional explanations.
>
> Your feedback is very important to us, and we sincerely appreciate the time and effort you have devoted to reviewing our work.
>
> Best regards,
>
> Authors

---

### Author Response · Authors · 2025-11-19
**Response to Reviewers' Comments and Summary of Revisions**

Dear Area Chairs and Reviewers,

We sincerely thank the reviewers for their time, effort, and constructive feedback. Their insightful comments have greatly strengthened the clarity and completeness of our work. We are encouraged by the positive assessments, which highlight the following strengths of our work:

- **Contribution**
  - **Addressing a long-standing evaluation gap.** The paper identifies overlooked limitations in current CIL evaluation protocols and brings attention to an underexplored but important direction (Reviewer Htje, Reviewer rrMk).
  - **A distributional perspective for CIL benchmarking.** Our work challenges reliance on a few random class orders and reframes evaluation as estimating full performance distributions (Reviewer Qscv, Reviewer SxFj).
  - **A simple and practical method.** EDGE is easy to implement and effectively uses CLIP-based label embeddings for most visual tasks (Reviewer Htje).
  - **Task similarity–difficulty connection.** The proposed link between similarity and difficulty provides an intuitive and useful new analytic perspective (Reviewer 6DZz).
- **Soundness**
  - **Clear theoretical grounding.** The paper offers rigorous analysis and theorems that quantify the limitations of random sampling (Reviewer Htje, Reviewer Qscv).
  - **Comprehensive experiments.** Both pretrained and non-pretrained CIL methods are systematically compared under RS and EDGE, giving deeper insights into evaluation robustness (Reviewer Htje, Reviewer Qscv).
- **Presentation**
  - **Clear writing.** The manuscript is well written and easy to follow (Reviewer rrMk, Reviewer Qscv).
  - **Emphasizing class-ordering effects.** The paper clearly illustrates how class ordering impacts evaluation reliability (Reviewer rrMk).
  - **Code availability.** Partial source code is included to support reproducibility (Reviewer 6DZz).

Several reviewers also raised important questions that helped us improve the paper. We have updated the appendix and discussion accordingly:

1. **Appendix E.3.2: Discussion on non-natural/professional images (pp. 30–31)**

   Reviewers Htje (W1, Q2), Qscv (W2, Q1) and SxFj (Q2) flagged concerns about CLIP’s reliability outside natural images. We added experiments on two medical imaging datasets and measured the agreement between task similarity matrices derived from class labels and from image prototypes.

   We **confirm** that directly using CLIP on raw class names can fail in domain-specific datasets. To mitigate this, we **propose a simple text-semantics augmentation** that expands class names into visually descriptive captions using a generative language model. After this quick augmentation, CLIP-based similarities better reflect image-based similarities, showing that our approach **can generalize to non-natural and professional domains**.

2. **Appendix E.4: Discussion of other potential baselines (pp. 31–32)**

   We thank Reviewers Htje (W2) and SxFj (W1) for suggesting more baselines. In response **we added four potential baselines**: single-round and multi-round LLM-generated sequences (LLM-1 and LLM-5), adversarial sampling (AS), and max-cover sampling (MCS). Details are provided in Appendix E.4 and results appear in Table A10.

   We find that these baselines outperform RS in some cases but face **two practical limitations:** lower efficiency and poor generalization of extreme sequences across the diverse set of CIL methods. Overall, EDGE remains the **most efficient** and **consistent method** for capturing extreme sequences across methods.

3. **Appendix E.5: Discussion on the number of samples (pp. 32-33)**

   To address concerns raised by Reviewer Htje (Q5) and Reviewer 6DZz (W1, Q1), we simulate settings with increased RS sample budgets. We run **two groups of experiments**, one in an enumerable setting and one in classic CIL settings.

   In the enumerable setting we observe that increasing RS samples improves its estimates, but RS typically requires about **twice the sample cost** to match the quality produced by three EDGE samples. If both RS and EDGE sample budgets increase, **EDGE remains superior until they converge at around 10–20 samples.**

   In classic CIL settings, we performed 600 RS samplings for L2P and 400 RS samplings for EASE. We observe that **RS needs hundreds of samples** on average to find extreme cases that EDGE captures in a single run. These experiments reinforce that EDGE offers substantially better sample efficiency and more reliable extreme-case detection than RS.

We deeply appreciate the reviewers' thoughtful feedback and constructive suggestions. In the following detailed responses, we address each comment individually and describe the revisions made in response to the reviewers’ recommendations.

Best regards,

Authors

---

### Comment · Reviewer_rrMk · 2025-11-25
**Fundamental Issue: How can this method be used in practice?**

I appreciate the clarification from the authors, but I am still not fully convinced about why this method would be useful in practice.

Models trained on the CIFAR dataset are rarely deployed in real settings. But let us assume the model was trained on the same data distribution that appears during deployment.

1. If we already know the class order before training, then why do we need continual learning at all? To use the introduced evaluation method, we must have access to future classes and future data. If we have this information, then we can simply train one model or update the continual learning model using the data from all classes.

2. If we do not know the class order and cannot control it, which is the most realistic situation, then even if we save the test data for all past tasks, we cannot use this evaluation method to judge whether the model is good, because in practice, we only have a single trained model.

I am trying to help the authors find an idea that is more practical and realistic. One possible approach could be to store a small subset of data from each task, similar to a buffer. Then we can train the model on the buffer using different sequences and identify which sequences are hard and which are easy. This seems closer to a real, continual learning scenario. However, it is still unclear whether a model trained only on the buffer can truly approximate the performance of the full model.

I would be happy to hear thoughts and insights from other reviewers as well.

---

> ### Author Response · Authors · 2025-11-26
> **Clarifying the Practical Role and Deployment Relevance of EDGE**
>
> Dear Reviewer rrMk,
>
> Thank you for your thoughtful critique and constructive suggestions. We address your concerns below in a Q&A format.
>
> **Q1:** Models trained on the CIFAR dataset are rarely deployed in real settings. But let us assume the model was trained on the same data distribution that appears during deployment.
>
> 1. If we already know the class order before training, then why do we need continual learning at all? To use the introduced evaluation method, we must have access to future classes and future data. If we have this information, then we can simply train one model or update the continual learning model using the data from all classes.
> 2. If we do not know the class order and cannot control it, which is the most realistic situation, then even if we save the test data for all past tasks, we cannot use this evaluation method to judge whether the model is good, because in practice, we only have a single trained model.
>
> **A1:** We believe there may be a **misunderstanding about the aim of CIL evaluation**. When a set of CIL algorithms is developed, two sequential stages typically follow:
>
> **[Step 1] Evaluation on benchmarks and model selection (the focus of this paper)**
>
> New CIL methods are conventionally **evaluated on benchmark datasets** such as CIFAR, CUB, and ImageNet, with results reported for accuracy, forgetting, and related metrics. **The purpose of this stage is to screen and select more reliable methods, thereby reducing risks in downstream deployment.**
>
> **Our paper addresses precisely this stage.** We show that the common practice of estimating performance by averaging over a few randomly sampled class orders can systematically misestimate the performance distribution and thereby increase deployment risk. EDGE is designed to operate in this controlled evaluation stage: it reveals method behavior under representative “hard” and “easy” orders and thus **helps researchers and practitioners make more reliable comparisons and choices.**
>
> **[Step 2] Deployment (real-world operation)**
>
> After selecting a robust method in Step 1, that method is deployed in the wild. In deployment, the class arrival order is **indeed uncontrollable**, and humans cannot intervene. **That is precisely why robust pre-deployment selection matters: the choice made in Step 1 directly affects safety and performance risk in deployment.**
>
> Therefore, there is **no paradox** that would require access to future classes at deployment. We do not use future data during deployment; instead, we perform rigorous, controlled pre-deployment evaluation to mitigate downstream risk. The two stages are **sequential and complementary**.
>
> **Q2:** I am trying to help the authors find an idea that is more practical and realistic. One possible approach could be to store a small subset of data from each task, similar to a buffer. Then we can train the model on the buffer using different sequences and identify which sequences are hard and which are easy. This seems closer to a real, continual learning scenario.
>
> **A2:** We appreciate your suggestion. However, we respectfully find the statement “we can train the model on the buffer using different sequences” **problematic**.
>
> (1) If your proposal applies to **[Step 1]**, then the experimental setup is essentially **the same as ours**. However, your approach appears to be **extremely costly** because it requires training the model across many different sequences to learn the relationship between sequence and performance. By contrast, EDGE requires no retraining. Inspired by continual learning theory, EDGE uses **only class labels** to quickly generate extreme sequences without performing actual training, and extensive experiments have validated EDGE’s effectiveness.
>
> (2) If your proposal applies to **[Step 2]**, then “using different sequences” **contradicts the deployment assumption** that the class arrival order is uncontrollable. In addition, performing such sequence-based evaluation in a live deployment environment would be **extremely risky**.
>
> In summary, thank you for your time and effort and for the valuable suggestion. We hope these clarifications address your concerns and we welcome further discussion.
>
> Sincerely,
>
> Authors

---

> > ### Comment · Reviewer_rrMk · 2025-11-26
> >
> > Thank you for clarifying that the focus of this work is on "Evaluation on benchmarks and model selection".
> >
> > 1. If that is the case, based on the results in Table 1, we would select the same model (RanPAC) irrespective of whether we use RS or EDGE. However, in Table 2, we would indeed select different models depending on the dataset, evaluation method (RS or EDGE), and difficulty level (easy region vs hard region).
> >
> > 2. Why is the ranking obtained in Table A5 and Table A6 inconsistent with the results in Table 1 and Table 2?
> >
> > 3. Let us consider that we want to select the model that does well in both easier and harder sequences. Could the authors show the result of ranking CL methods in Tables 1 and 2 using RS and EDGE, but we use the mean performance of the hard and easy sequences obtained with RS and EDGE?
> >
> > Additional Comment:
> > In the author's comment above: "By contrast, EDGE requires no retraining. Inspired by continual learning theory, EDGE uses only class labels to quickly generate extreme sequences without performing actual training,", does not EDGE still require training on easy and hard sequences after identifying the sequences?

---

> > > ### Author Response · Authors · 2025-12-01
> > > **Response and Clarifications on Model Selection**
> > >
> > > Dear Reviewer rrMk,
> > >
> > > Thank you for your response. We address your comments below in a Q&A format.
> > >
> > > **Q1:** If that is the case, based on the results in Table 1, we would select the same model (RanPAC) irrespective of whether we use RS or EDGE. However, in Table 2, we would indeed select different models depending on the dataset, evaluation method (RS or EDGE), and difficulty level (easy region vs hard region).
> > >
> > > **A1:** **Your observation is correct, and this is precisely the point we aim to highlight.** EDGE provides an additional and meaningful perspective for model selection: one may select models based on lower-bound performance, upper-bound performance, or robustness. We **do not claim** that RS is entirely wrong. For methods like RanPAC, which are inherently robust to task-order variations, RS already produces reliable estimates. For other methods, however, **EDGE serves as a valuable complement by revealing order-sensitive behaviors that RS may overlook.**
> > >
> > > **Q2:** Why is the ranking obtained in Table A5 and Table A6 inconsistent with the results in Table 1 and Table 2?
> > >
> > > **A2:** Thank you for your careful inspection. This discrepancy arises because these tables come from **different experimental settings**, which we have now **clarified and revised** in **Appendix E.1.3**. Specifically, Table 1 and Table 2 follow the experimental protocol described in Section 5.1 (using a fixed set of the first six classes). In contrast, the tables in the appendix use a *randomly selected* group of six classes to provide an additional robustness sanity check. These are two independent experiments, and their rankings may differ. Nonetheless, **both consistently support our key conclusion**: EDGE provides rankings that more closely reflect true performance across task-order variability.
> > >
> > > **Q3:** Let us consider that we want to select the model that does well in both easier and harder sequences. Could the authors show the result of ranking CL methods in Tables 1 and 2 using RS and EDGE, but we use the mean performance of the hard and easy sequences obtained with RS and EDGE?
> > >
> > > **A3:** While this is possible, we consider it **unnecessary and potentially misleading**. Collapsing the hard and easy results into a single average score tends to **mask the very performance characteristics that matter for model selection**. For instance, two methods may yield identical averages yet possess markedly different risk behaviors: Method $\mathbf{A}$ (Hard=70, Easy=90, mean=80) is substantially more stable than Method $\mathbf{B}$ (Hard=60, Easy=100, mean=80), whose performance varies widely across sequences. Although the mean coincides, the associated reliability is fundamentally different. **Using a single averaged metric would therefore eliminate precisely the distinctions that EDGE is designed to reveal.**
> > >
> > > This is why we deliberately report three complementary dimensions:
> > >
> > > - the lower bound, which reflects worst-case performance and is crucial in safety-critical settings
> > > - the upper bound, which reflects the best attainable performance
> > > - stability/variance, which captures sensitivity to task-order fluctuations
> > >
> > > Together, these dimensions **provide a richer basis for decision-making than a single average could.**
> > >
> > > **Q4:** Additional Comment: In the author's comment above: "By contrast, EDGE requires no retraining. Inspired by continual learning theory, EDGE uses only class labels to quickly generate extreme sequences without performing actual training,", does not EDGE still require training on easy and hard sequences after identifying the sequences?
> > >
> > > **A4:** This point may have been interpreted differently than we intended. The statement refers specifically to the **sequence generation** stage. EDGE **does not** require training any model in order to search for extreme sequences. Once sequences are generated, however, **every evaluation protocol** including RS and EDGE **must train** the corresponding continual learning models on those sequences in order to **report final accuracies**.
> > >
> > > We hope these clarifications address your concerns.
> > >
> > > Sincerely,
> > >
> > > Authors

---

### Meta-Review · Area_Chair_Semw · 2026-01-07

**Summary:**

- This paper shows the major limitations of the existing evaluation protocol of class incremental learning  and proposes a new evaluation framework to address this issue. Both theoretical analysis and empirical experiments demonstrate the better estimation of the performance by using the new evaluation framework.

- Five reviews were collected. Three reviewers were positive: 8, 6, 6. One reviewer who gave a score of 4 decided to raise the score to 6 during the discussion. The reviewer with the most negative opinion ( Reviewer rrMk with a score of 4) criticizes the practicability of the proposed method.

- While I partially agree with the practicability issues raised by Reviewer rrMk, these issues are fundamental limitations of the class incremental learning (CIL) framework, instead of issues unique to the proposed method. This paper can be viewed as one step towards making the classical CIL framework more realistic, but it would be unreasonable to expect all such limitations to be resolved by one paper.

- Therefore, the strengths of this paper outweigh the weaknesses and thus lead to an Accept.

**Reviewer Concerns:**

**Resolved concerns**
- Three reviewers raised the concerns about the reliability of CLIP outside natural images. The rebuttal added additional experiments on medical imaging datasets and proposed a simple augmentation to mitigate the limitations of CLIP.

- Reviewers Htje and SxFj suggested more baselines. The rebuttal added four baselines to the experiments.

- Reviewers Htje and Reviewer 6DZz questions the increase in the sample costs. The rebuttal provided additional experiments to demonstrate the sample efficiency of the proposed method.


**Unresolved concerns**
Reviewer rrMk questions the practicability of the proposed method and the limitations of the CIL framework. These are important questions to be addressed in the future.

**Reviewer Scores:**

Five reviews were collected. Three reviewers were positive: 8, 6, 6. I don't think they will change the score. One reviewer who gave a score of 4 decided to raise the score to 6 during the discussion. The reviewer with the most negative opinion ( Reviewer rrMk with a score of 4) is not likely to change the score.

---

### Decision · Program_Chairs · 2026-01-26

Accept (Poster)